# TGFβ suppresses CD8$^+$ T cell expression of CXCR3 and tumor trafficking

Andrew J. Gunderson[1], Tomoko Yamazaki[1], Kayla McCarty[1], Nathaniel Fox[1], Michaela Phillips[1], Alejandro Alice[1], Tiffany Blair[1], Mark Whiteford[1,2], David O'Brien[1,2], Rehan Ahmad[1,2], Maria X. Kiely[1,2], Amanda Hayman[1,2], Todd Crocenzi[1], Michael J. Gough[1], Marka R. Crittenden[1,3] & Kristina H. Young[1,3 ✉]

Transforming growth factor beta (TGFβ) is a multipotent immunosuppressive cytokine. TGFβ excludes immune cells from tumors, and TGFβ inhibition improves the efficacy of cytotoxic and immune therapies. Using preclinical colorectal cancer models in cell type-conditional TGFβ receptor I (ALK5) knockout mice, we interrogate this mechanism. Tumor growth delay and radiation response are unchanged in animals with Treg or macrophage-specific ALK5 deletion. However, CD8αCre-ALK5$^{flox/flox}$ (ALK5$^{ΔCD8}$) mice reject tumors in high proportions, dependent on CD8$^+$ T cells. ALK5$^{ΔCD8}$ mice have more tumor-infiltrating effector CD8$^+$ T cells, with more cytotoxic capacity. ALK5-deficient CD8$^+$ T cells exhibit increased CXCR3 expression and enhanced migration towards CXCL10. TGFβ reduces CXCR3 expression, and increases binding of Smad2 to the CXCR3 promoter. In vivo CXCR3 blockade partially abrogates the survival advantage of an ALK5$^{ΔCD8}$ host. These data demonstrate a mechanism of TGFβ immunosuppression through inhibition of CXCR3 in CD8$^+$ T cells, thereby limiting their trafficking into tumors.

[1] Earle A. Chiles Research Institute, Providence Cancer Institute, 4805 NE Glisan St, Portland, OR 97213, USA. [2] The Oregon Clinic, Colon and Rectal Surgery Division, 4805 NE Glisan St, Suite 6N60, Portland, OR 97213, USA. [3] The Oregon Clinic, Radiation Oncology Division, 4805 NE Glisan St, G level, Portland, OR 97213, USA. ✉email: Kristina.Young@providence.org

Transforming growth factor beta (TGFβ) is a multipotent cytokine with complex roles in tumorigenesis including epithelial to mesenchymal transition[1], angiogenesis[2], tumor cell motility and metastasis[3], cancer associated fibroblast (CAF) proliferation[4], and immunosuppression[5]. TGFβ exists in its latent form in the tumor microenvironment, and can be activated through multiple mechanisms (reviewed in[6]), including radiation. Once activated, dimeric TGFβ binds the heteromeric receptor consisting of two copies of the type I receptor, TGFβRI (or Activin receptor Like Kinase 5 (ALK5)), and two copies of the type II receptor (TGFβRII). Binding of TGFβ to TGFβRII leads to phosphorylation of the serine/threonine kinase, ALK5, which to phosphorylation of the intracellular signaling mediators, Mothers against decapentaplegic homolog 2 and 3 (Smad2 and Smad3), which are then capable of binding the common Smad, Smad4, leading to nuclear translocation of the Smad complex. Once in the nucleus, Smad3 and Smad4 bind DNA at Smad-binding elements, termed CAGA boxes, and regulate transcription of TGFβ target genes. TGFβ is known to suppress T cell effector function, in part, through Smad-mediated downregulation of the target genes granzyme, perforin, and interferon[5]. As we and others have previously shown, blockade of TGFβ signaling improves response to cytotoxic therapies, depends upon adaptive immunity, and synergizes with immune checkpoint blockade[7–11]. Furthermore, the detection of a TGFβ gene expression signature correlates with T cell exclusion from tumors and resistance to immunotherapy[12–14]. Although these studies have demonstrated the association of anti-TGFβ therapies with enhanced T cell infiltration and anti-tumor immunity, the mechanism by which TGFβ excludes immune cells and limits the efficacy of immune therapies is unknown. Herein, we propose a mechanism to clarify this critical immunosuppressive role of TGFβ.

Chemokine receptor CXC motif 3 (CXCR3) is binds and traffics towards its IFNγ-inducible ligands, CXCL9, 10, and 11. It is primarily expressed on activated CD8+ T cells, NK cells and CD4+ $T_H1$ cells with critical roles in recruiting and retaining T cells during infection, autoimmunity, and cancer[15]. Previous work in cancer immunity has demonstrated that CXCR3 is necessary for trafficking and efficacy of adoptively transferred anti-tumor T cells[16,17], as well as mediating tumor regression following anti-PD1 therapy[18,19]. In colorectal cancer, active secretion of CXCL10 is associated with granzyme B-expressing CD8+ T cell infiltration and more favorable TNM staging[20]. Furthermore, treatment of tumor bearing animals with myeloid targeting agents enhanced CXCR3-mediated tumor homing resulting in improved survival[21,22], highlighting the critical role of CXCR3 mediated T cell chemotaxis as a common pathway to effective anti-tumor immunity.

In this study, we investigate how TGFβ in conjunction with cytotoxic therapy suppresses anti-tumor immunity through the use of clinical ALK5 small molecule inhibitors and cell-type conditional ALK5-deficient mice. Blockade of ALK5 phosphorylation prior to chemo-radiation treatment significantly reduces tumor growth and extends survival. This is associated with and dependent upon increased CD8+ T cell tumor infiltration. The predominant therapeutic effect of ALK5 inhibition is directly on CD8+ T cells; CD8α-specific deletion of ALK5 enhances CXCR3 expression on CD8+ T cells resulting in increased CXCR3-dependent migration into tumors. CXCR3 is directly suppressed by SMAD2/3 downstream of TGFβ. Once in the tumor microenvironment, ALK5-deficient T cells exhibit a decreased threshold for T cell receptor (TCR) activation and cytotoxicity. These data demonstrate a mechanism by which TGFβ contributes to immunosuppression through down-regulation of CD8+ T cell expression of CXCR3, limiting trafficking to the tumor. These findings demonstrate a mechanism by which TGFβ contributes to immune suppression that can be targeted in clinical trials.

## Results

**TGFβ inhibition sensitizes tumors to chemoradiation.** As we and others have previously shown[7,11,23,24], TGFβ blockade combined with radiation reduces tumor growth in various murine models. We tested this therapeutic strategy using a clinically relevant small molecule inhibitor of ALK5, LY2157299 (LY, a.k.a. Galunisertib), administered prior to chemo-radiation in mice bearing established colorectal tumors. We evaluated LY in combination with 5-fluorouracil (5-FU) chemotherapy and radiation (RT), either 2 Gy × 15 ($BED_{10}$ 36) or 5 Gy daily for 5 consecutive days ($BED_{10}$ 37.5), mirroring standard of care clinical dosing schedules for neoadjuvant treatment of rectal cancer patients (Fig. 1a), which demonstrated equivalent efficacy (Supplementary Fig. 1a). Although RT + 5FU alone provided a modest survival advantage over vehicle control (median survival 25d vs. 23d, $p < 0.05$; Fig. 1b), the addition of LY significantly slowed tumor growth (Fig. 1bi, $p < 0.01$ at day 13 and day 23), and provided the greatest survival benefit to tumor-bearing mice (Fig. 1Bii, median survival 41d, $p < 0.05$ vs. RT + 5-FU, $p < 0.0001$ vs. vehicle). In order to understand whether adaptive immunity contributed to this therapeutic effect, we depleted CD4+ cells and CD8α+ cells prior to treatment, beginning day 4. For practicality, we used 5 Gy × 5 for the depletion studies. Our data demonstrate CD8α+ cells were required for efficacy, but CD4+ cellular depletion prior to chemoradiation improved the efficacy over RT + 5FU alone and recapitulated the RT + 5FU + LY efficacy, however it did not provide any additional benefit when added to RT + 5FU + LY (Fig. 1c). These data indicate that CD8α+ cells, but not CD4+ cells, are necessary for the improved efficacy of chemoradiation plus ALK5 inhibition in CT26 tumors. Evaluation of tumor sections harvested from mice at day 14 (one day post-LY treatment) demonstrated reduced phosphorylation and nuclear translocation of the TGFβ signaling mediator, Smad2, indicating ALK5 signaling was attenuated in tumor tissue, specifically in the CD8α+ cells (Supplementary Fig. 1b). The slight improvement in tumor control with CD4+ T cell depletion over RT + 5FU alone (Fig. 1c) suggests that CD4+ T regulatory cells (Treg) may also be a target of LY leading to less inhibition of CD8α+ T cells, or may be a source of the TGFβ that inhibits CD8α+ T cells, such that CD4-depletion abrogates the immunosuppressive effects. We subsequently evaluated production of TGFβ by cells within the tumor. Tregs, which expressed the highest baseline levels of latent TGFβ1/latency associate peptide (LAP) protein, also increased in frequency following RT (Supplementary Fig. 1c, d); however, TGFβ was expressed by all cell types. Neutrophil and macrophage production of TGFβ increased significantly following radiation consistent with their role in wound healing and phagocytosis following tumor cell apoptosis (Supplementary Fig. 1c)[25,26].

**CD8α+ T cells are the direct target of TGFβ inhibitor.** To clarify further the primary target of LY2157299, we utilized the Cre-Lox system to generate double transgenic mice via cell-type specific Cre expression. We employed Lyz2-Cre[27] (monocytes/macrophages), Foxp3-CreERT2-eGFP[28] (regulatory T cells), and CD8α-Cre[29] (CD8α+ T cells) animals, and crossed them with ALK5flox/flox mice[30] to excise exon 3 of the ALK5 gene. Double transgenic mice demonstrated specific ALK5 excision by PCR evaluation of flow cytometry isolated immune cells from tumors and spleens (Supplementary Fig. 2a, b). These animals were subsequently challenged with syngeneic colorectal MC38 tumors, as all transgenic animals shared the C57BL/6 background. Tumors took uniformly in ALK5$^{ΔLyz2}$ animals, and tumor growth

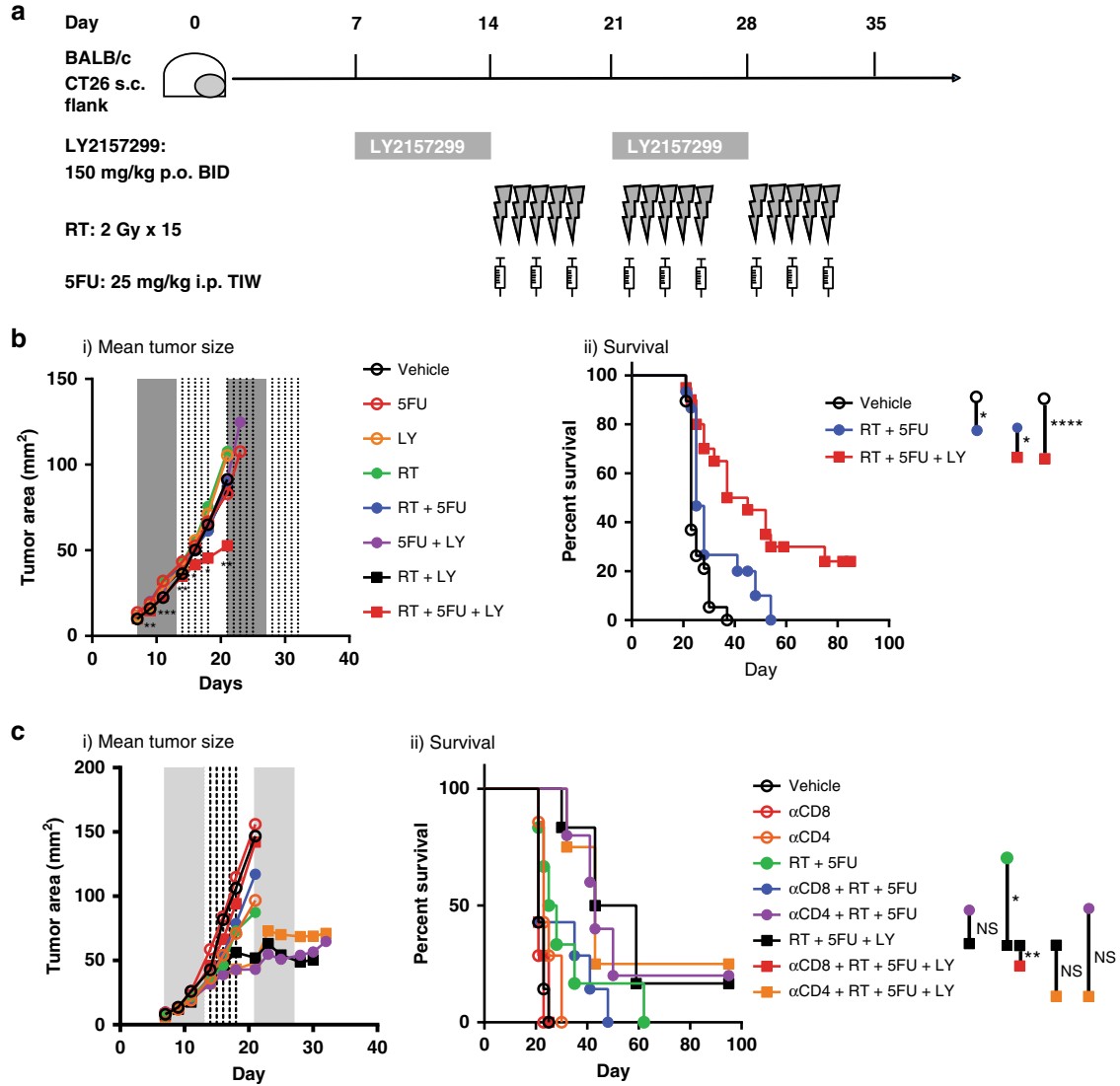

**Fig. 1 Enhancement of chemoradiation efficacy by the TGFβRI inhibitor, LY2157299, depends on CD8α⁺ cells in CT26 tumors. a** Treatment schema in the CT26 tumor model. Six to eight-week-old BALB/c mice were implanted subcutaneously in the left flank with $2 \times 10^5$ CT26 tumor cells. Seven days following implant animals were randomized based on tumor size; vehicle control or LY2157299 (LY) compound were administered by oral gavage twice daily every 12 h at 150 mg/kg for 7 consecutive days. At day 14, some mice received 15 radiation treatments of 2 Gy, delivered Monday through Friday, targeted to tumor tissue along with 25 mg/kg injections of 5-fluorouracil (5FU) chemotherapy administered intraperitoneally on days 14, 16, and 18. LY treatment groups resumed BID oral dosing from days 21–27. **b** Mean tumor growth (i) and survival (ii) measured 3 times per week up to 100 days or death. For (i), p-values are a comparison between RT + 5FU (blue dots) and RT + 5FU + LY (red squares) using an unpaired, two-tailed *t*-test, and values are as follows for days 7–21: 0.059 (NS), 0.0086 (**), 0.0001(***), 0.0048 (**), 0.039(*), 0.042(*), 0.0068(**). For (ii), *p*-values for survival were derived using log-rank test. Vehicle (open circle) vs. RT + 5FU = 0.023(*), vehicle vs. RT + 5FU + LY < 0.0001(****), RT + 5FU vs. RT + 5FU + LY = 0.015(*). Displayed is a combined results of 3 independent experiments with *n* = 15 mice/group. **c** Mean tumor size (i) and survival (ii) of CT26 tumor-bearing mice receiving the indicated treatment combinations, with RT dose of 5 Gy × 5 consecutive fractions, and either anti-CD4 or anti-CD8α monoclonal antibodies to deplete CD4⁺ and CD8α⁺ cells prior to therapy. The legend key for each group is located next to the survival curve. NS = not significant, *p = 0.0448, **p = 0.0011. *P*-value was derived using the Wilcoxon test. N = 6 mice/group. Displayed is one representative experiment of 2 total experiments.

and survival were similar to C57BL/6 J controls (Fig. 2a). There was a non-significant increase in cured animals following radiation in the ALK5$^{\Delta Lyz2}$ animals compared to control (0% vs. 20% cure rate, *p* = 0.2 by Fischer's exact).

To our surprise, there was more rapid tumor growth in ALK5$^{\Delta Foxp3}$ animals, but no difference in survival or radiation response (Fig. 2b, 31 vs. 55 mm² at day 16 (v), *p* < 0.05). A previous publication utilizing the ALK5$^{\Delta Foxp3}$ mice found FoxP3⁺ Tregs of the colonic lamina propria were better able to suppress CD8⁺ T cell IFN-γ production when *ALK5* was lost due to enhanced Treg expression of the transcription factor Tbet[31].

Therefore, to determine if tumor infiltrating Tregs harbored a similar, more suppressive phenotype, we evaluated regulatory T cell Tbet expression in MC38 tumors. More tumor-infiltrating Foxp3⁺ Tregs expressed Tbet in ALK5$^{\Delta Foxp3}$ mice compared to littermate control (LM) (Supplementary Fig. 2c), suggesting a more suppressive regulatory T cell phenotype in ALK5$^{\Delta Foxp3}$ mice may be contributing to the more rapid tumor growth.

MC38 tumors grew to comparable sizes by 10–14 days post implant in ALK5$^{\Delta CD8}$ and wildtype (WT) animals (Fig. 2c), however, tumors were subsequently rejected in >60% of ALK5$^{\Delta CD8}$ transgenic animals (Fig. 2c). This translated to

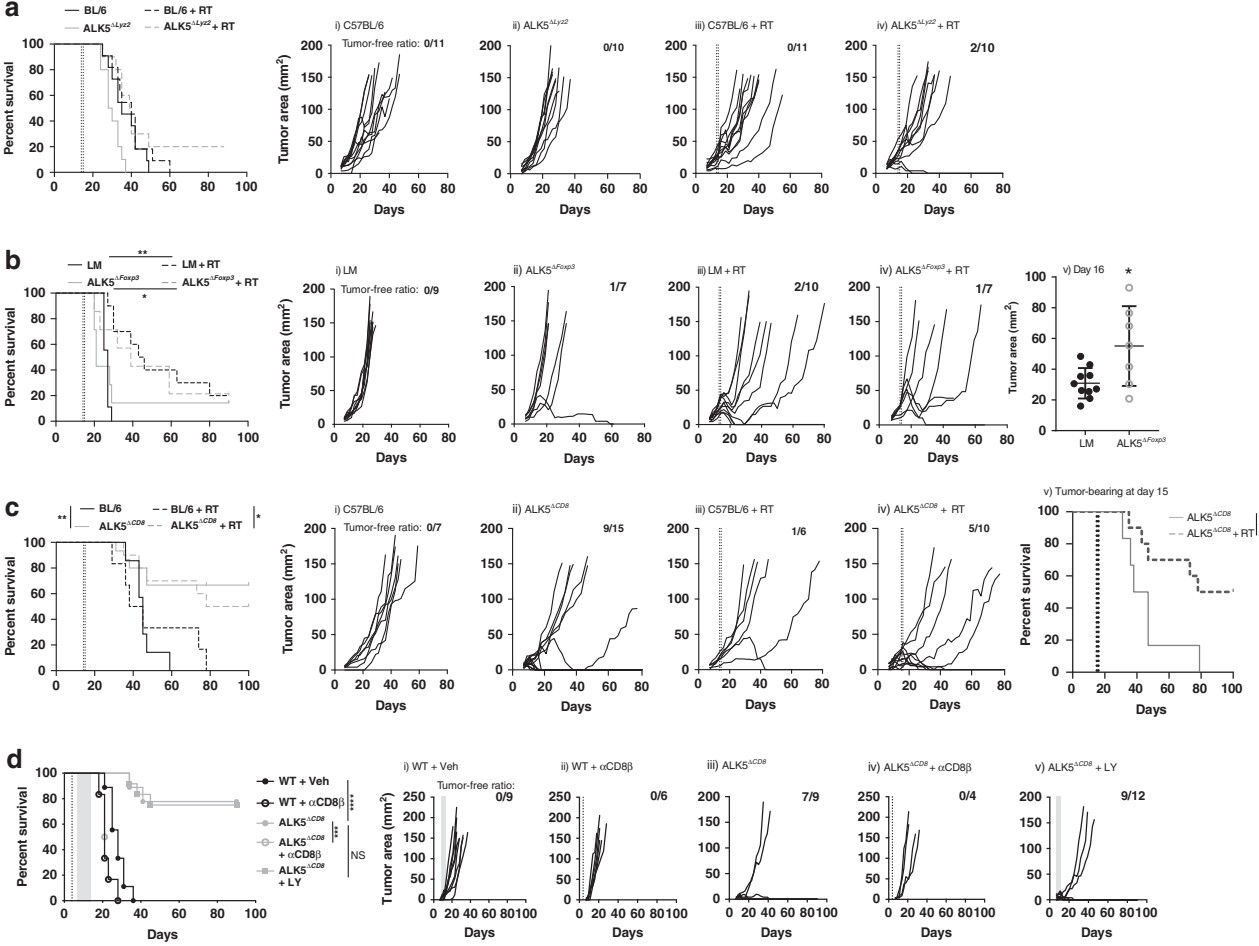

**Fig. 2 CD8α+ cells are a direct target of TGFβRI mediated immune suppression.** MC38 tumor-bearing animals underwent radiation in 2 consecutive doses of 10 Gy when tumors were 25 mm². Mice were euthanized when tumors reached 144 mm². **a** Survival curves for C57BL/6 ($n = 11$), Lyz2Cre-ALK5$^{flox/flox}$ (ALK5$^{\Delta Lyz2}$) ($n = 10$), C57BL/6 + RT($n = 11$), and ALK5$^{\Delta Lyz2}$ ($n = 10$) animals (left), (i–iv) Individual tumor growth curves. P-values were derived using log-rank test. Shown is the compilation of 2 independent experiments. **b** Tumor growth and survival of littermate control (LM) ($n = 9$), Foxp3-eGFP-CreERT2/ALK5$^{flox/flox}$ (ALK5$^{\Delta Foxp3}$) ($n = 7$), LM + RT ($n = 10$), and ALK5$^{\Delta Foxp3}$ ($n = 7$) mice. Prior to tumor implantation, all mice received 5 consecutive daily i.p. injections (1 mg/injection) of tamoxifen emulsified in sunflower oil. (i–iv) Individual tumor growth curves. Shown is 1 representative experiment reflective of 2 independent experiments. P-values were derived using log-rank test. *$p = 0.0196$, **$p = 0.0004$. v) Tumor area presented as mean +/− SD at day 16 in LM ($n = 10$) vs. ALK5$^{\Delta Foxp3}$ mice ($n = 7$). Statistical comparison with unpaired, two-tailed t-test. **c** Survival and individual growth curves of MC38 tumor-bearing mice in C57BL/6 ($n = 7$), CD8aCre-ALK5$^{flox/flox}$ (ALK5$^{\Delta CD8}$) ($n = 15$), C57BL/6 + RT ($n = 6$), ALK5$^{\Delta CD8}$ ($n = 10$) mice. Shown is 1 representative experiment of 3 independent experiments. (i–iv) Individual tumor growth curves. P-values were derived using log-rank test. *$p = 0.0265$, **$p = 0.0047$. (v) Survival curve for ALK5$^{\Delta CD8}$ animals who failed to reject their tumors by day 15, and were subsequently randomized +/− radiation; ALK5$^{\Delta CD8}$ ($n = 6$) and ALK5$^{\Delta CD8}$ + RT ($n = 10$). P-values were derived using log-rank test. *$p = 0.0216$. **d** C57BL/6 and ALK5$^{\Delta CD8}$ mice bearing MC38 tumors treated with anti-CD8β mAb on day 4. LY was administered via oral gavage twice daily (150 mg/kg) for 7 days. N as follows: WT + Veh=9, WT + aCD8β ($n = 6$), ALK5$^{\Delta CD8}$ = 9, ALK5$^{\Delta CD8}$ + aCD8β = 4, ALK5$^{\Delta CD8}$ + LY = 12. P-values were derived using log-rank test ***$p = 0.0002$, ****$p < 0.0001$, NS = not significant. (i–v) Individual tumor growth curves. Data from 1 representative experiment is shown reflective of 2 independent experiments. The number of mice cured over the total number of tumor-bearing mice is shown in the top right of each graph.

improved survival of ALK5$^{\Delta CD8}$ mice (median survival not reached vs. 45 days in WT mice, $p < 0.01$, Fig. 2c). We next sought to evaluate whether the effects of radiation were different in WT vs. ALK5$^{\Delta CD8}$ mice. When mice were randomized at day 14 to receive radiation, all tumors under 25 mm² in ALK5$^{\Delta CD8}$ mice treated with RT were eradicated. However, given the high rate of tumor rejection in ALK5$^{\Delta CD8}$ mice it was difficult to assess the radiation effect. Therefore, to better assess the response to radiation in ALK5$^{\Delta CD8}$ mice, it was necessary to select for animals whose tumors were not rejected, presumably a more aggressive, immunosuppressed phenotype. We waited until day 15, when it was clear that tumors would take, then randomized ALK5$^{\Delta CD8}$ mice to hypofractionated radiation (10 Gy × 2). Radiation significantly improved survival of ALK5$^{\Delta CD8}$ animals

compared to ALK5$^{\Delta CD8}$ mice who failed to reject tumors by day 15 (Fig. 2Cv, median survival 42.5d vs. 89d, $p < 0.05$). Radiation in ALK5$^{\Delta CD8}$ mice was more effective than in WT control (median survival 89d vs. 41.5d, $p < 0.05$, Fig. 2c). In addition, we observed a non-significant increase in cure rates among ALK5$^{\Delta CD8}$ compared to WT animals receiving radiation, 50% vs. 13.6% in WT mice (Fig. 2a–c, $p = 0.18$ by $\chi^2$). Thus, CD8α-specific loss of *ALK5* results in higher rates of tumor rejection, improved survival, and enhanced response to radiation.

We next evaluated whether the improved survival and radiosensitivity observed in ALK5$^{\Delta CD8}$ mice was dependent on CD8+ T cells. MC38 tumor-bearing mice were treated with an anti-CD8β antibody on day 4, which depletes CD8+ T cells, but not CD8α-expressing dendritic cells (Supplementary Fig. 3a).

ALK5$^{\Delta CD8}$ mice treated with anti-CD8β grew tumors with similar kinetics and survival as wildtype control mice (median survival 24.5d vs. 28d, $p = 0.24$, Fig. 2d). These data demonstrate that CD8$^+$ T cells are necessary for the improved survival and enhanced tumor rejection observed in ALK5$^{\Delta CD8}$ mice.

In order to evaluate whether the improved efficacy of RT + 5FU + LY (Fig. 1b) was due to the direct effect of ALK5 inhibition on CD8$^+$ T cells, we tested LY treatment in ALK5$^{\Delta CD8}$ mice. There was no improvement in survival or tumor growth kinetics with the addition of LY2157299 (Fig. 2d). These data suggest the primary target of LY2157299 is the CD8$^+$ T cell, via inhibition of ALK5. This is significant, as it has been reported that LY2157299 has a lower Kd for ALK4 than ALK5, raising the possibility that bone morphogenic protein (BMP) signaling through ALK4 may have contributed to the efficacy observed with RT + 5FU + LY therapy[32]. To further demonstrate that ALK5 inhibition is the primary mechanism for efficacy, we tested a more selective second generation ALK5 inhibitor, LY3200882[33]. Using this more potent ALK5 inhibitor with chemoradiation, we observed greater efficacy than was seen with LY2157299, achieving cures in 6 of 7 animals (median survival not reached vs. 28d RT + 5FU, $p < 0.001$, Supplementary Fig. 3b, c). Further, cured animals rejected tumor rechallenge at day 73 post-implant with CT26 cells but not the immunologically distinct 4T1 cell line implanted simultaneously on the opposite flank (Supplementary Fig. 3d), demonstrating the generation of tumor-specific immune memory. Together, these data demonstrate that LY2157299 acts primarily via ALK5 inhibition of CD8$^+$ T cells.

**TGFβ increases the activation threshold of CD8$^+$ T cells**. We evaluated the mechanism by which CD8 T cell-specific ALK5 loss improved anti-tumor immunity by seeking differences in immune cell populations in the periphery and the tumor microenvironment of ALK5$^{\Delta CD8}$ mice. Flow cytometric evaluation of MC38 tumors derived from ALK5$^{\Delta CD8}$ or WT mice at day 14 post implant (prior to rejection) revealed an increase in total CD8$^+$ T cells in tumors and a corresponding decrease in CD8$^+$ T cells in tumor draining lymph nodes from ALK5$^{\Delta CD8}$ mice (Fig. 3a). However, no difference was observed in the percentage of CD8$^+$ T cells reactive against the MC38 trackable antigen p15E (Fig. 3b). We next evaluated whether CD8$^+$ T-cell subsets varied between tumors in ALK5$^{\Delta CD8}$ and WT animals. In ALK5$^{\Delta CD8}$ mice there was a decrease in tumor infiltrating naïve CD8$^+$ T cells (CD44$^{LO}$/CD62L$^+$), and an increase in effector CD8$^+$ T cells (CD44$^{INT}$CD62L$^-$) (Fig. 3c), also defined by Ly6C$^+$ CD62L$^-$(Fig. 3d)[34,35]. To determine whether ALK5 loss contributed to the development of a novel T cell subset or enriched an existing subset, we generated a t-Distributed Stochastic Neighbor Embedding (tSNE) plot from our flow cytometry staining markers (Supplementary Fig. 4a). Using next-nearest neighbor clustering, we observed an enrichment of the effector population without generation of a novel subset of tumor-infiltrating T cells (Supplementary Fig. 4a). No significant differences were observed in CD4$^+$ T cells or T helper conventional and regulatory T cell subsets in tumors or lymph nodes (Supplementary Fig. 4b). Macrophage frequency was reduced in tumors from ALK5$^{\Delta CD8}$ mice (Supplementary Fig. 4c).

We next evaluated the function of infiltrating CD8$^+$ T cells in WT and ALK5$^{\Delta CD8}$ animals. The percent of tumor infiltrating CD8$^+$ cells that expressed IFN-γ or TNF-α were similar in ALK5$^{\Delta CD8}$ and WT mice (Supplementary Fig. 4d). However, within the effector CD8$^+$ T cell subset, there was an increase in granzyme B expression (Fig. 3e), consistent with previous reports of TGFβ-mediated transcriptional suppression of granzyme[5]. We subsequently tested tumor-specific cytotoxicity of ALK5$^{\Delta CD8}$ and

WT CD8$^+$ T cells ex vivo. Ovalbumin (OVA)-specific CD8$^+$ T cells were generated by vaccination with a replication deficient L. monocytogenes, engineered to express the OVA peptide SIINFEKL. CD8$^+$ T cells were isolated from splenocytes and co-cultured at increasing ratios of effector to tumor cell, with an OVA-expressing tumor cell line (MCA205-OVA) or a control Panc02 tumor cell line, which does not express OVA but is derived from C57BL/6 mice (Fig. 3f). We observed enhanced tumor specific cytotoxicity of ALK5$^{\Delta CD8}$ CD8$^+$ T cells compared to WT control, and no difference in non-specific cytotoxicity (Fig. 3f). A recent publication demonstrated that ALK5 expression in CD4$^+$ T cells is regulated by TCR signal strength[36]. We therefore interrogated whether the reciprocal was true in CD8$^+$ T cells; we tested whether ALK5 loss altered the threshold for TCR stimulation. Splenocyte-derived, purified naïve CD8$^+$ T cells were cultured with a fixed concentration of αCD28 antibody and increasing amounts of plate-bound agonist αCD3 antibody (Fig. 3g). We observed increased proliferation and production of IFNγ and TNFα at lower concentrations of αCD3 antibody suggesting that loss of ALK5 decreased the threshold for TCR-mediated CD8$^+$ T cell activation. These data indicate that TGFβ suppresses anti-tumor CD8$^+$ T cell function by raising the threshold for naïve T cell activation through TCR stimulation, resulting in decreased effector differentiation and cytotoxicity.

LY2157299, which inhibits TGFβ signaling, altered CD8 T cell function and tumor infiltrating immune cells similar to what was observed in tumors from ALK5$^{\Delta CD8}$ animals. There were more CD8$^+$ T cells and fewer macrophages and Tregs infiltrating tumors from LY-treated animals (Supplementary Fig. 4e), consistent with changes observed in ALK5$^{\Delta CD8}$ animals. There was also an increase in the percentage of CD8$^+$ T cells expressing the T-Box transcription factors EOMES and Tbet from spleens, lymph nodes and CT26 tumors of LY-treated mice consistent with enhanced CD8$^+$ T cell effector differentiation (Supplementary Fig. 4f). Together these data suggest that ALK5 inhibition with LY2157299 is capable of generating an increase in tumor-infiltrating effector CD8$^+$ T cells.

**TGFβ suppresses CXCR3 and CD8$^+$ T cell tumor infiltration**. Based on the detection of increased tumor-infiltrating CD8$^+$ T cells in ALK5$^{\Delta CD8}$ and LY-treated mice, we hypothesized that TGFβ may inhibit either in situ proliferation or tumor trafficking of CD8$^+$ T cells, or both. To evaluate these possibilities simultaneously, we adoptively co-transferred CFSE-labeled congenic WT and ALK5$^{\Delta CD8}$ CD8$^+$ T cells, at a 50:50 ratio, into WT mice with established day 14 MC38 tumors. Seven days following transfer, spleens, tumor draining lymph nodes, and tumors were harvested for flow cytometric analysis (Fig. 4a). A significantly higher percentage of WT cells could be detected in the spleen and lymph nodes, with a significantly greater proportion of ALK5$^{\Delta CD8}$ T cells in tumors (Fig. 4b). To determine whether the increased infiltration of ALK5$^{\Delta CD8}$ T cells into tumors was due to a proliferative advantage, CFSE reduction was assessed in the infiltrating cells. There was no significant difference in CFSE labeling between tumor-infiltrating WT vs. ALK5$^{\Delta CD8}$ CD8$^+$ T cells, though increased proliferation of ALK5$^{\Delta CD8}$ CD8$^+$ T cells was observed in the spleen and lymph node (Fig. 4c and Supplementary Fig. 5a). Consistent with these findings, the percentage of Ki67$^+$CD8$^+$ T cells was increased in the draining lymph nodes of ALK5$^{\Delta CD8}$ mice (Supplementary Fig. 5b). As TGFβ has a known role in suppressing proliferation, we tested whether ALK5$^{\Delta CD8}$ CD8$^+$ T cells had a proliferative advantage ex vivo. Naïve splenocytes were cultured in vitro with TGFβ1 with and without CD3/CD28 stimulation and evaluated for proliferation. TGFβ-mediated suppression of proliferation was observed in WT

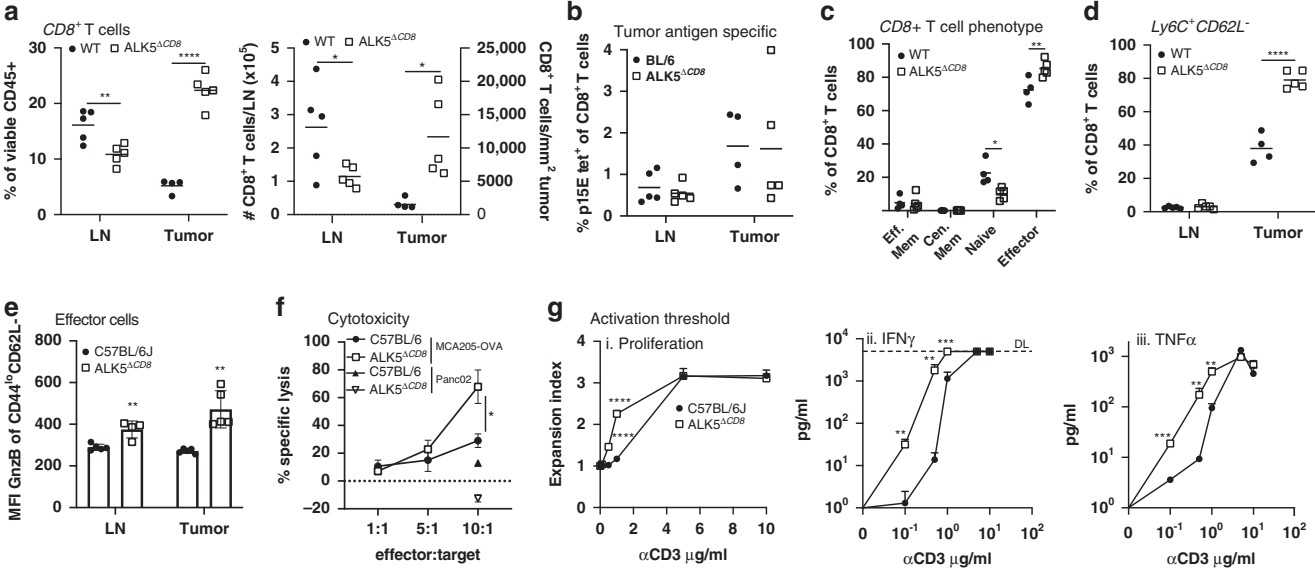

**Fig. 3 ALK5 loss lowers the TCR activation threshold and enhances effector function of CD8$^+$ T cells.** For **a–e**: On day 14 following MC38 implantation into WT ($n = 6$) and ALK5$^{\Delta CD8}$ ($n = 5$) mice, tumors were harvested, digested into a single cell suspension, and analyzed by flow cytometry. Each symbol is the value of an individual animal, measure of center is the mean. **a** Quantification of CD8$^+$ T cells in lymph nodes and tumors by frequency (left). **$p = 0.00765$, ****$p = 0.000013$. Absolute cell number in the lymph node or per mm$^2$ in the tumor (right). *$p = 0.043$ (LN) and 0.0172 (tumor). **b** The frequency of p15E tetramer positive CD8$^+$ T cells determined by FACS analysis in lymph nodes and tumors of WT and ALK5$^{\Delta CD8}$ mice. **c** The frequency of effector memory (CD44$^{hi}$/CD62L$^-$), central memory (CD44$^{hi}$/CD62L$^+$), naïve (CD44$^{lo}$/CD62L$^+$) or effector (CD44$^{lo}$/CD62L$^-$) CD8$^+$ T cells. *$p = 0.01$, **$p = 0.0092$. **d** The frequency of Ly6C$^+$CD62L$^-$ CD8$^+$ T cells in MC38 tumors or tumor draining inguinal lymph nodes from C57BL/6 or ALK5$^{\Delta CD8}$ mice. ****$p = 0.000046$. **e** The median fluorescent intensity of granzyme B (GnzB) expression gated on CD44$^{lo}$CD62L$^-$CD8$^+$ T cells in MC38 tumors, presented as mean +/− SD. **$p = 0.00316$ (LN) and 0.0011 (tumor). **f** Specific cytotoxic killing of tumor cells was measured with the Incucyte live cell imaging system in co-cultures of MCA205-OVA or Panc02 tumor cells with CFSE-labeled C57BL/6 WT and ALK5$^{\Delta CD8}$ CD8$^+$ T cells isolated from spleens of ∆actA-OVA vaccinated mice. Shown is the cumulative % dead tumor cells as determined by cytotox red reagent uptake of total tumor cells seeded following 24 h of co-culture presented as mean +/− SD. $N = 3$/condition. *$p = 0.0344$. **g** CD8$^+$ T cells purified from spleens of naïve WT and ALK5$^{\Delta CD8}$ mice and activated with increasing amounts of plate bound anti-CD3 and 10μg/ml anti-CD28. Presented as mean +/− SD. $N = 3$/condition (i) Proliferation index measured by CFSE dye dilution, ****$p = 0.000015$ and 0.000043. (ii) IFN-γ secretion, **$p = 0.00658$ (10$^{-1}$), 0.00924 (10$^0$), ***$p = 0.00016$, and (iii) TNF-α secretion **$p = 0.00828$ (10$^0$), 0.00182 (10$^1$), ***$p = 0.00096$ (10$^{-1}$). Each graph is reflective of two independent experiments. $P$-values derived using unpaired, two-tailed $t$-test, except in **g**, which utilized 1-way ANOVA with multiple comparisons.

CD8$^+$T cells, but not in the ALK5$^{\Delta CD8}$ CD8$^+$ T cells (Fig. 4d). Taken together, these data suggest: (a) proliferation may be suppressed by means other than TGFβ in the tumor microenvironment, and (b) improved tumor trafficking was responsible for the increased CD8$^+$ T cell infiltration.

Therefore, we proceeded to evaluate mechanisms of increased tumor trafficking that could be attributed to changes in the CD8$^+$ T cells harboring *ALK5* deletion. Given the enhanced cytotoxicity of ALK5$^{\Delta CD8}$ CD8$^+$ T cells and diminished macrophage infiltrate into tumors, we first evaluated for differences in cytokines and chemokines from digested tumors grown in WT and ALK5$^{\Delta CD8}$ animals by multiplex cytokine bead array. We observed minimal differences in cytokine levels in the tumor of WT and ALK5$^{\Delta CD8}$ mice (Supplementary Fig. S4g), consistent with altered levels of macrophage infiltrate. To determine whether the expression of chemokine receptors on CD8 T cells could explain the differential infiltration of WT and ALK5$^{\Delta CD8}$ T cells, we evaluated CXCR3 and CXCR6, which are dominant chemokine receptors for CD8$^+$ T cell trafficking into tumors[37], particularly following radiation[38,39]. We observed an increase in CXCR3 expression in transferred ALK5$^{\Delta CD8}$ CD8$^+$ T cells in the co-transfer assay (Fig. 4e and Supplementary Fig. 5c), but not CXCR6 (Supplementary Fig. 5d). We then tested whether CXCR3 expression was altered by TGFβ treatment ex vivo. Splenocyte-derived CD8$^+$ T cells from WT animals demonstrated an increase in CXCR3 expression with CD3/CD28 stimulation, which was inhibited by TGFβ1 (Fig. 4f and Supplementary Fig. 5e). However, CD8$^+$

T cells from ALK5$^{\Delta CD8}$ animals demonstrated a very high baseline expression of CXCR3, which was minimally decreased by aCD3/CD28 stimulation, while TGFβ1 treatment had no effect (Fig. 4f and Supplementary Fig. 5e). Consistent with increased CXCR3 expression, an increase in ALK5$^{\Delta CD8}$ CD8$^+$ T cell in vitro migration towards CXCR3 ligand CXCL10 was observed compared to WT in a dose dependent manner (Fig. 4g and Supplementary Fig. 5f), but migration towards CXCR6 ligand CXCL16 was not observed in either WT or ALK5$^{\Delta CD8}$ CD8$^+$ T cells (Supplementary Fig. 5f). CXCL10 protein levels were equivalent between tumors implanted in ALK5$^{\Delta CD8}$ and WT animals at day 14 in the MC38 tumor model (Supplementary Fig. 4g). We therefore interpret the enhanced CD8$^+$ T cell trafficking to be the result of modulation of CXCR3 expression on CD8$^+$ T cells.

In order to assess whether CXCR3 was a direct transcriptional target of TGFβ in T cells, we performed chromatin immunoprecipitation for the TGFβ signaling mediators Smad2 and Smad3, and performed qPCR of the CXCR3 promoter region identified to contain Smad-binding elements, up to 5000 bp upstream of the transcriptional start site. The human CXCR3 promoter in the human Jurkat cell line exhibited a significant increase in Smad2 and Smad3 binding approximately 4000 bp upstream of the transcriptional start site, 1.5 h after TGFβ treatment (Fig. 5a). We similarly observed increased in Smad2 binding the murine CXCR3 promoter approximately 3800 bp upstream of the TSS in splenocyte-derived purified CD8$^+$ T cells following TGFβ

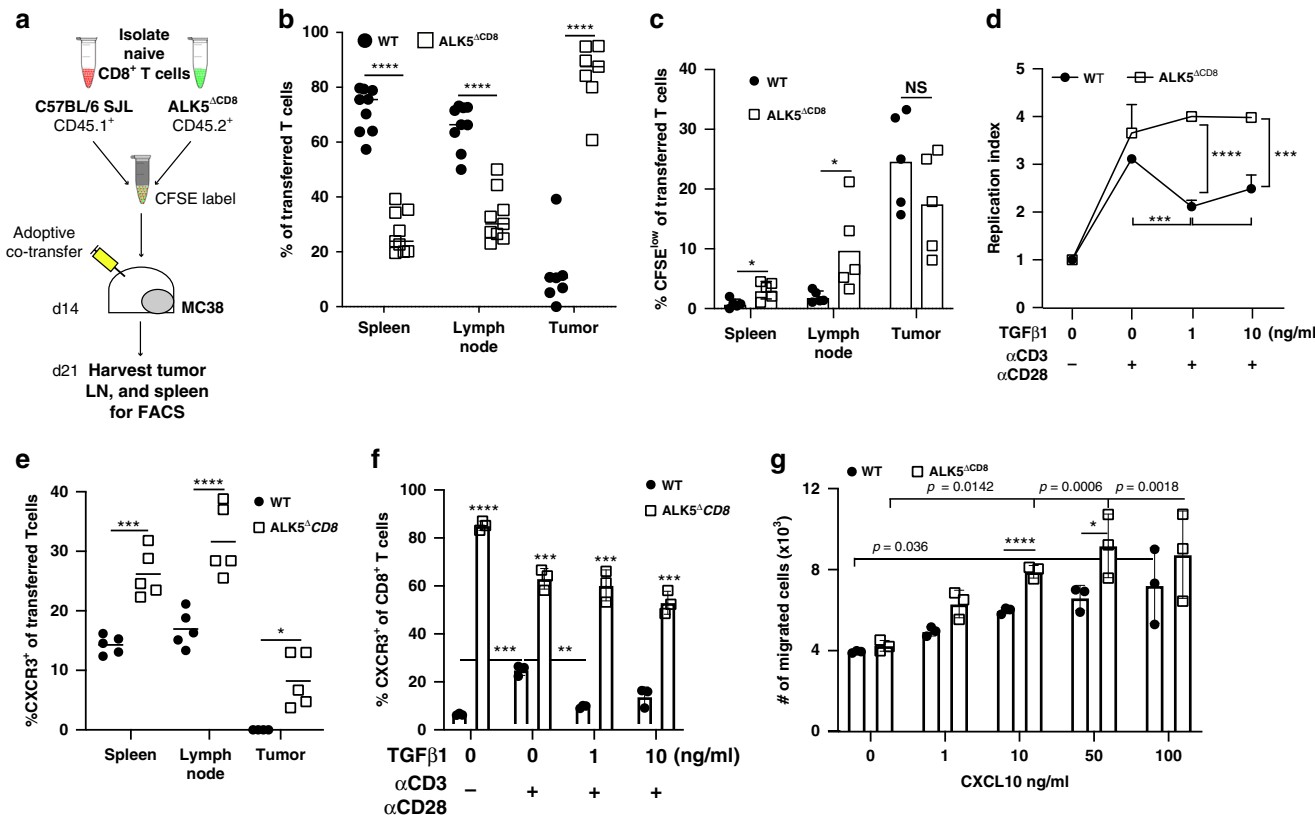

**Fig. 4 ALK5$^{\Delta CD8}$ T cells upregulate CXCR3 and preferentially migrate to the tumor. a** Experimental schema for adoptive co-transfer of congenic C57BL/6 WT and ALK5$^{\Delta CD8}$ CD8$^+$ T cells into MC38 tumor-bearing mice. **b** The frequency of transferred CD45.1 WT and CD45.2 ALK5$^{\Delta CD8}$ CD8$^+$ T cells ($n = 9$ recipients) analyzed in the spleen, lymph node, and tumor; each symbol represents one mouse, measure of center = mean, CD45.1 WT transferred T cells (black circles) and CD45.2 ALK5$^{\Delta CD8}$ transferred T cells (white boxes). Shown is a compilation of two independent experiments. ****$p < 0.000001$. **c** The percent of CFSE$^{low}$ cells gated on CD45.1 WT or CD45.2 ALK5$^{\Delta CD8}$ transferred T cells in spleen, lymph node, and MC38 tumors. *$p = 0.0155$ (spleen), 0.04393 (lymph node). $N = 5$ recipients, reflective of two independent experiments. **d** In vitro proliferation of naïve splenocyte-derived CD8$^+$ T cells from C57BL/6 or ALK5$^{\Delta CD8}$ mice stimulated with plate-bound αCD3/αCD28 +/− TGFβ1 (1 or 10 ng/ml) for 68 h; $n = 3$ biologic replicates/group, displayed as mean + SD. One representative experiment is shown reflective of 3 independent experiments. P-values as follows: WT vs. ALK5$^{\Delta CD8}$ = 0.000016 (1 ng/ml), 0.00086 (10 ng/ml); WT vs. WT = 0.0002 (0 vs. 1 ng/ml), 0.0201 (0 vs. 10 ng/ml). **e** The percent of CXCR3$^+$ transferred CD8$^+$ T cells assessed by FACS analysis 7 days post transfer in tumor bearing mice. Symbols represent one mouse, measure of center=mean, $N = 5$ recipients, reflective of two independent experiments. *$p = 0.011$, ***$p = 0.0002$, ****$p = 0.000083$. **f** Frequency of CXCR3$^+$CD8$^+$ T cells in vitro following treatment as in **d**. $n = 3$ biologic replicates/group. One experiment is shown reflective of 3 independent experiments, displayed as mean +/− SD. For WT vs. ALK5$^{\Delta CD8}$, ****$p < 0.000001$ (unstimulated); ***$p = 0.000173$ (0 ng/ml TGFβ); ***$p = 0.000175$ (1 ng/ml); ***$p = 0.00042$ (10 ng/ml). For WT vs. WT, ***$p = 0.0003$ (unstimulated vs. 0 ng/mL); **$p = 0.0023$ (0 vs. 1 ng/mL). **g** In vitro migration of WT and ALK5$^{\Delta CD8}$ CD8$^+$ T cells to CXCL10 in the bottom chamber, displayed as mean +/−SD, $n = 3$ biologic replicates/group, reflective of 2 independent experiments. WT vs. ALK5$^{\Delta CD8}$: $p = 0.00067$ (10 ng/mL); $p = 0.0489$ (50 ng/mL). For panels **d**, **f**, and **g**, $p$-values were derived using 1-way ANOVA with multiple comparisons, all other panels used unpaired, two-tailed $t$-test.

stimulation (Fig. 5b), while Smad3 bound the promoter constitutively and did not increase with TGFβ stimulation (Fig. 5b). Our interpretation of these data is that CXCR3 is transcriptionally repressed by TGFβ, leading to decreased CXCR3 expression on CD8$^+$ T cells and impaired chemotaxis to its ligands. To determine the clinical relevance of our data, we interrogated the expression of CXCR3 in patients. Analysis of the TCGA colorectal database revealed an inverse correlation between SMAD2 and CXCR3 expression (Fig. 5c), consistent with our data implicating TGFβ as a suppressor of CXCR3 expression.

To test whether enhanced CXCR3 expression is necessary for the rejection of MC38 tumors and extended survival observed in tumor-bearing ALK5$^{\Delta CD8}$ mice, we blocked CXCR3 in vivo with an anti-CXCR3 blocking antibody delivered on days 4, 8, and 12 after tumor challenge (Fig. 5d, e). The majority of ALK5$^{\Delta CD8}$ mice rejected MC38 tumors, whereas all tumors progressed in

WT animals (Fig. 5d). However, anti-CXCR3 interfered with the anti-tumor activity, with 83% of ALK5$^{\Delta CD8}$ mice treated with αCXCR3 antibody developing tumors (Fig. 5di, iv). Consistent with this, tumor infiltration by CD8$^+$ T cells was significantly reduced by administration of αCXCR3 antibody (Fig. 5e). The tumors that developed in the ALK5$^{\Delta CD8}$ mice grew with delayed growth kinetics, likely attributed to the enhanced effector function of ALK5$^{\Delta CD8}$ CD8$^+$ T cells that infiltrated the tumors, albeit at lower frequencies after CXCR3 blockade.

We then asked if CXCR3-dependent chemotaxis could be increased by utilizing an ALK5 inhibitor. In our preclinical modeling, we observed an increase in CXCR3$^+$CD8$^+$ T cells following treatment with LY2157299 (Supplementary Fig. 5g), indicating increased CXCR3 expression was not a developmental artifact of ALK5$^{\Delta CD8}$ transgenic mice. Of note, the frequency of tumor-infiltrating CXCR3$^+$CD8$^+$ T cells steadily decreases over time, with complete loss of CXCR3$^+$ cells 21 days post

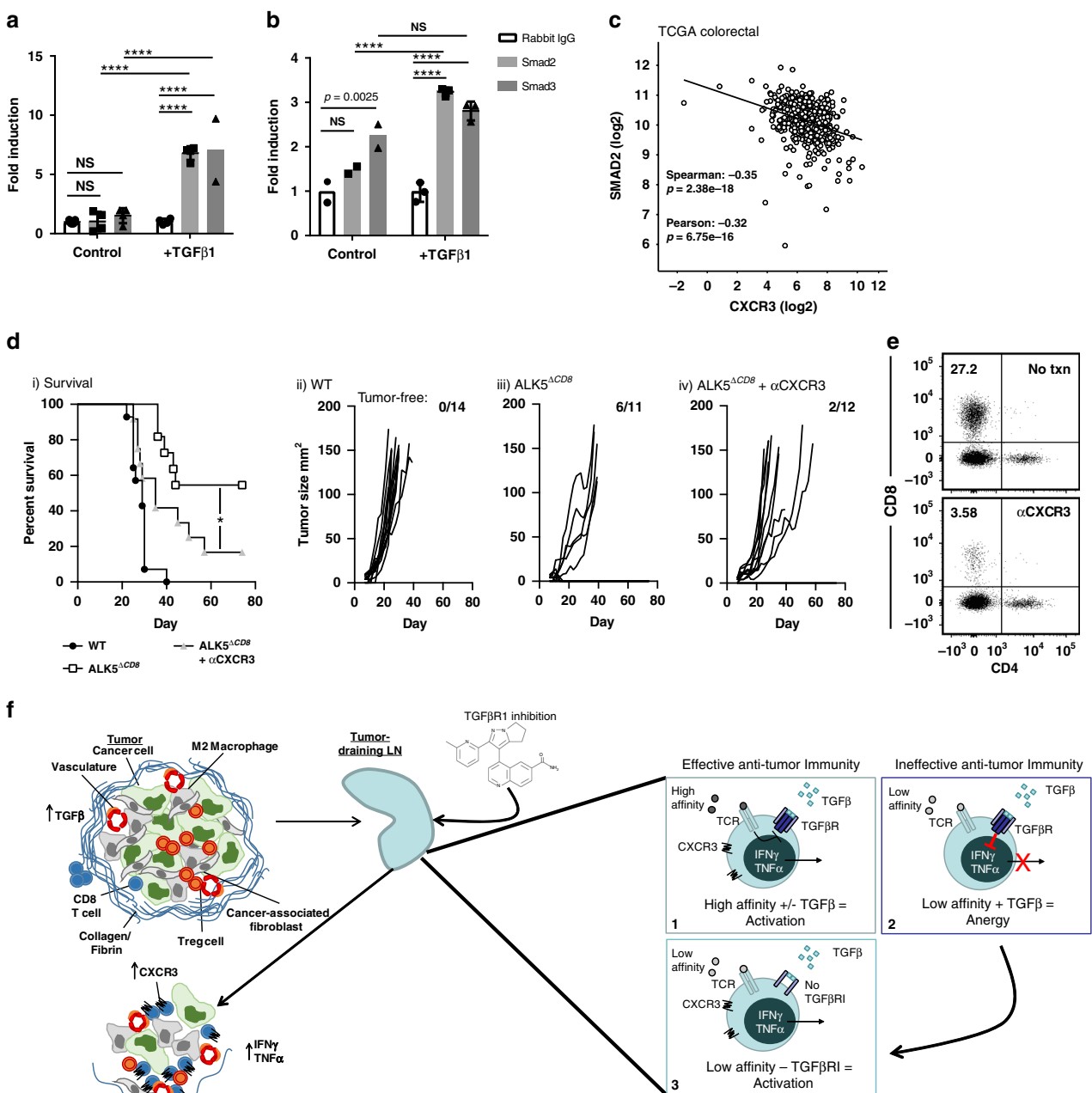

**Fig. 5 TGFβ-mediated suppression of CXCR3 limits CD8+ T cell chemotaxis and anti-tumor effect.** ChIP was carried out using human Jurkat cells (**a**) and mouse primary CD8+ T cells (**b**) with anti-SMAD2 and anti-SMAD3 antibodies +/− TGFβ1 treatment, $n = 2$–5 technical replicates/condition. Experiment shown is representative of 3 independent experiments, displayed as mean +/− SD. ****$p < 0.0001$ using one-way ANOVA with Sidak's multiple comparison correction. **c** Linear regression analysis of SMAD2 and CXCR3 gene expression levels from a cohort of TCGA colorectal cancer patients. *P*-values derived from Spearman and Pearson correlations. **d** MC38 survival (i) and tumor growth curves (ii–iv) in WT ($n = 14$), ALK5$^{ΔCD8}$ ($n = 11$), and ALK5$^{ΔCD8}$ + αCXCR3 blocking antibody ($n = 12$). The ratio of cured/total mice in each group is in the upper right. Shown is a composite of two independent experiments. *P*-value derived from log-rank test. *$p = 0.0388$. **e** Representative FACS plots of tumors from one C57BL/6 mouse each from untreated or αCXCR3 treated groups harvested 4 days after the last dose. Shown is CD4 and CD8 expression gated on viable CD45+CD3+ cells. **f** Schematic representing hypothesis for LY2157299 mechanism of action. Naïve T cells exposed to LY are more susceptible to activation by low-avidity antigen in the lymph node. CXCR3 is upregulated upon activation, and by loss of TGFβ-mediated suppression, which allows for better homing to CXCR3 ligands produced in the tumor microenvironment endogenously and following radiation. Once in the tumor, those TGFβ-resistant CD8 T cells have improved cytotoxicity.

implantation in MC38 (Supplementary Fig. 5h), which may reflect the development of an exhausted phenotype where CXCR3 is known to be downregulated[40]. These data demonstrate that inhibition of TGFβ with a small molecule inhibitor can increase CXCR3 expression.

Taken together, these data demonstrate a mechanism by which the response to cytotoxic therapy can be improved: TGFβ acts locally to suppress the transcription of CXCR3 thereby limiting tumor-infiltration; CD8+ T cells that do reach the tumor have an increased threshold for activation, decreased cytotoxicity, and

decreased proliferation. These data demonstrate that TGFβ inhibition is capable of altering chemokine receptor expression on T cells to promote chemotaxis to the tumor, and improve response to chemoradiation.

## Discussion

This study establishes a mechanism for TGFβ-mediated immunosuppression. We report that TGFβ signaling can directly regulate expression of a chemokine receptor in T cells. Previous studies revealed that TGFβ signaling promoted T cell exclusion and lymphocyte sequestration at tumor margins, thereby contributing to resistance to T cell mediated immune therapies and promotion of tumor metastases[14,41]. Herein, we demonstrate repression of CXCR3 expression is one mechanism for TGFβ-mediated exclusion of T cell trafficking to tumors. This has broad applicability for improving the efficacy of CD8+ T cell mediated immunotherapies that require T cell infiltration into tumors. For example, reports that elevated TGFβ-associated gene signatures are linked to resistance to immune checkpoint blockade[13] may be due, in part, to reduced trafficking to the tumor through CXCR3 repression. This provides further mechanistic rationale for the synergy reported between TGFβ inhibition and immune checkpoint blockade[10,24,42,43]. Our data demonstrate that the increase in T cell trafficking into poorly infiltrated tumors following TGFβ blockade may rely on tumor and/or tumor microenvironment expression of CXCR3 ligands: CXCL9, CXCL10, or CXCL11. Radiation can stimulate the CXCR3 chemokines CXCL9, 10 and 11, following high dose radiation[44], and this dependent on type 1 and type 2 interferons[45,46] providing a means of enhancing CXCR3-mediated T cell trafficking to the tumor. Therefore, one mechanism by which radiation and TGFβ inhibition synergize is via radiation-mediated upregulation of CXCR3 ligands, resulting in CXCR3-dependent recruitment of effector T cells. CXCR3 ligand expression can cooperate with vascular cell adhesion molecules to mediate homing of CD8+ T cells to malignant tissue[47], and suppression of CXCR3 ligands is an immune escape mechanism for B16 lung metastasis in mice[48]. Furthermore, blockade of TGFβ signaling is likely to enhance other T cell directed therapies, such as adoptive T cell transfer which is dependent on CXCR3 chemotaxis[16,49].

Upon infiltration into the tumor, T cells resistant to TGFβ demonstrate increased cytotoxicity and expression of effector differentiation markers including Ly6C+, CD62L−, and granzyme B+. Interestingly, no difference was observed in the frequency of tumor-antigen specific p15E+CD8+ T cells. This is in contrast to previous data demonstrating rejection of EL4 and B16 tumor cells in CD4-dnTGFβRII animals[50]. These mice express a dominant negative TGFβRII in both CD4+ and CD8+ T cell compartments, whereas our ALK5ΔCD8 mice restricts TGFβRI deletion to the CD8+ T cells. Interestingly, CD4-dnTGFβRII and CD4-TGFβRII-KO animals do not phenocopy each other, and the ALK5ΔCD8 mice more closely resemble the CD4-TGFβRII-KO strain[51]. These discrepancies raise the possibility that the dominant negative TGFβRII may act as a partial TGFβ trap absorbing free TGFβ in the microenvironment providing broader inhibition of TGFβ across multiple cell types, in addition to cell-specific ablation of signaling. Despite the similar frequency of antigen-specific CD8 T cells, we still observed (1) increased numbers of infiltrating T cells leading to increased absolute counts of p15E+ cells, and (2) increased cytotoxicity on a per-cell basis, even amongst p15E-negative infiltrating CD8+ T cells. Our model is limited in our ability to assess for lower affinity antigen-specific T cells, which we hypothesize may be increased in ALK5ΔCD8 animals given that the threshold for TCR activation was decreased by TGFβR blockade (Fig. 3). New anti-tumor immune responses may be generated during TGFβ blockade by lower affinity antigen-TCR interactions. In addition, strong TCR stimulation via high concentrations of αCD3 resulted in maximal proliferation and cytokine production, and abrogated any differential advantage observed in the ALK5ΔCD8 CD8+ T cells (Fig. 3g). These data suggest TCR stimulation by a high affinity tumor antigen, such as p15E, would generate similarly functional anti-tumor efficacy in WT vs. ALK5ΔCD8 animals (i.e., at maximum threshold); while lower affinity p15E-negative CD8+ effector T cells may be the population that is enriched following TGFβ blockade, and responsible for the improvement in anti-tumor efficacy observed. Based on these data, we propose a model whereby the dominant effect of LY is in both the tumor and the tumor draining lymph node (Fig. 5f), such that naïve T cells exposed to LY have impaired TGFβ signaling rendering them more susceptible to activation by low-affinity antigen in the lymph node. CXCR3 is upregulated upon activation, and by loss of TGFβ-mediated suppression, which allows for better homing to CXCR3 ligands produced in the tumor microenvironment endogenously and enhanced following radiation. Once in the tumor, those TGFβ-resistant CD8 T cells have improved cytotoxicity. Low TGFβ-regulated extracellular matrix signatures have been shown to associate with improved response to immune checkpoint inhibitors[13], this may be also be contributing to enhanced anti-tumor immunity.

Because of the roles of TGFβ in wound healing and regulatory T cell development, we anticipated knockout of ALK5 in macrophages and Tregs to alter survival and radiation sensitivity. However, our data failed to demonstrate a benefit to TGFβ blockade in monocytes/macrophages and regulatory T cells in our model. Further, we demonstrate that tumor infiltrating Tregs deficient in ALK5 exhibit increased Tbet expression, known to be associated with enhanced immunosuppressive function[31]. Though there are many higher orders of post-translation regulation of TGFβ (reviewed in[52]), the improvement in survival observed with CD4+ T cell depletion (Fig. 1) suggests that Treg-derived TGFβ suppresses CD8+ T cell function. These results may be context and/or tumor-type dependent, as leukocyte content and phenotype can have significant effects on treatment response and tumor outcome[53]. Although they are not the primary target of TGFβRI inhibition in these models, macrophages and Tregs are major sources of TGFβ production in the MC38 tumor microenvironment. Numerous studies have shown that macrophage-derived TGFβ is necessary for tumor cell invasiveness and metastatic spread[54], though this demonstrates that monocyte-specific TGFβ signaling may reduce the efficacy of radiation. The CD4 depletion results in the context of increased CXCR3 recruitment following TGFβ blockade support a concept that Tregs may be suppressing CD8+ T cells via TGFβ. CXCR3 neutralization demonstrated only partial reversal of tumor control in ALK5ΔCD8 animals, so this may be one mechanism by which TGFβ is active in these models.

It is still unknown how SMAD2/3 acts as a repressor on the CXCR3 promoter. Tbet is necessary to upregulate CXCR3 on both CD4+ and CD8+ T cells[55,56] and we observed upregulation of Tbet following TGFβR1 blockade. Tbet expression can be regulated by TGFβ1 in T cells, but this appears to indirect[57–59]. Thomas and Massague reported that TGFβ directly inhibits cytotoxic effector gene expression following activation of CD8+ T cells. This effect is dependent on SMAD2/3 and ATF1/CREB binding of GnzB, Perforin, and IFNγ promoter regions[60]. We also demonstrated that ALK5-deficient CD8+ T cells were superior cytotoxic effectors in vivo and in vitro compared to wild-type cells (Fig. 5) potentially due to the interruption of TGFβR signaling that inhibits T cell differentiation[61,62]. We observed an alteration in the ratio of naïve-to-effector cells residing in the

tumor (Fig. 3c), suggesting TGFβ is limiting effector cell differentiation. Others have suggested CXCR3 expression favors short-lived effectors at the expense of memory T cell development[63,64]. However, we observed no change in the frequency of memory T cells (Fig. 3), and demonstrated adequate generation of tumor-specific immunity by resistance to re-challenge after tumor eradication with chemoradiation + LY therapy (Supplementary Fig. 3). These data indicate that the development of adaptive memory and recall responses remain intact in the absence of TGFβ signaling. Other reports demonstrate CD8+ T cell effector function and CXCR3 expression may be independently regulated as Tbet is not required for IFN-γ expression[65]. Similarly, in a model of adoptive T cell transfer, CXCR3 expression was necessary for tumor control, but not for cytotoxic T cell function[16]. Loss of CXCR3 expression can lead to decreased effector function indirectly, due to decreased migration to inflamed sites and therefore decreased interaction with antigen presenting cells[18]. Therefore, upregulation of CXCR3 in ALK5-deficient CD8+ T cells may indirectly reinforce effector phenotypes[63]. Interestingly, we observed loss of CXCR3 expression in tumor-infiltrating CD8+ T cells over time, with complete loss by day 21, which may reflect the development of exhaustion, as has been shown during chronic LCMV infection[40,66]. We did observe loss of CXCR3 in tumor-infiltrating CD8+ T cells from both WT and ALK5$^{\Delta CD8}$ animals suggesting that exhaustion is occurring through TGFβ-independent mechanisms, and may provide an opportunity for synergy with immune therapies targeting exhaustion, such as immune checkpoint blockade.

We are conducting a clinical trial of LY2157299 in combination with standard of care fractionated radiotherapy and 5-FU or capecitabine chemotherapy prior to surgery for patients with locally advanced rectal cancer (Clinicaltrials.gov identifier NCT02688712 and Fig. 5b). The purpose of these studies was to clarify the mechanism by which this therapy may be effective using a treatment schema in mice that reflected the clinical study (Fig. 1a). Since the initiation of the clinical trial and conduction of these studies, a 2nd generation ALK5 inhibitor has been developed, LY3200882, which is more specific and potent than the LY2157299 compound that was primarily used throughout this study. We anticipate, based on our genetic knockout models and our tumor growth and survival studies with the LY3200882 compound, that this newer drug will be more effective than the LY2157299 compound, and that the mechanism of action will be consistent with our data included herein. The primary endpoint of our clinical trial is the rate of pathologic complete responses. Pathologic complete response following neoadjuvant chemoradiation for rectal cancer is a predictor of both decreased local relapse and improved survival[67,68]. Intensification of neoadjuvant therapy for rectal cancer can improve the pathologic complete response rate[69]. This study is still accruing, so we await the final results to determine whether the addition of TGFβR1 inhibition significantly improves pathologic responses. Our data demonstrate TGFβ suppression of CXCR3 can be overcome with LY2157299, resulting in improved CD8+ T cell migration into tumors. We hypothesize based on our preclinical data this may improve response to neoadjuvant treatment in rectal cancer patients.

## Methods

**Animal studies and cell culture**. All experiments were approved by our institutional IACUC under protocol #54, and performed in our OLAW certified animal facility (Assurance #D16-00526). C57BL/6, BALB/c, CD8Cre, Lyz2Cre, and Foxp3-eGFP-CreERT2 mice were purchased from the Jackson Laboratories (Bar Harbor, Maine). ALK5$^{flox/flox}$ mice were a generous gift from Andrew Weinberg (Earle A. Chiles Research Institute). All transgenic mice were on C57BL/6 background. CT26, MC38, 4T1 and MCA205-OVA tumor cells were grown in DMEM medium supplemented with 10% heat inactivated fetal bovine serum (FBS) and 1%

penicillin-streptomycin (Pen/Strep) to 60–90% confluence prior to tumor implantation. All cells were washed 2× with 1× PBS and implanted subcutaneously in a 100 μl volume of 1× PBS in the lower flanks. $1 \times 10^5$ MC38 cells, $2 \times 10^5$ CT26 cells, $5 \times 10^4$ 4T1 cells and $5 \times 10^6$ MCA205-OVA cells were used for tumor induction. Tumors were measured 3×/week until they reached the survival end-point of 144 mm². LY2157299 and LY3200882 were provided via a materials transfer agreement with Eli Lilly. Mice were dosed via twice daily oral gavage at 105–150 mg/kg with doses spaced 12 h apart for the indicated durations. 5-fluorouracil (5-FU) chemotherapy was administered at 25 mg/kg through intraperitoneal (i.p.) injections three times per week for one week. Radiation was delivered with the Small Animal Radiation Research Platform (SARRP, Xstrahl, Atlanta, GA) with cone beam-CT image guidance with isocenter located centrally within the tumor mass, treated at a 45-degree angle with the indicated doses calculated using tissue-density from segmentation of the CT. Anti-CD4 (GK1.5), anti-CD8α (2.43), anti-CD8β (53–5.8), and anti-CXCR3 (CXCR3-173) depleting or neutralizing antibodies were purchased from BioXcell and i.p. injected at 200 μg/ mouse at the indicated time-points in the figures. Spleen, lymph nodes and tumors were harvested from animals and single cell suspensions were prepared using mechanical disaggregation for spleen and lymph nodes only or mincing and enzymatic digestion for 30 min. at 37 °C for tumors and lymph nodes for dendritic cell evaluation. Enzyme digest buffer included 1 mg/ml Collagenase A (Roche), 1 mg/ml Hyaluronidase (Sigma) and 50 u/ml DNase (Roche) in DMEM serum free base medium. Following single cell preparation, cells were washed and resuspended in FACS buffer (1× PBS, 1% BSA, 2 mM EDTA) and counted prior to FACS staining using a Guava EasyCyte cytometer (Millipore). Primary T cells were cultured in complete RPMI media (10% heat inactivated FBS, 1% Na-P, 1% NEAA, 10 mM HEPES, 55 μM β-mercaptoethanol, and 1% Pen/Strep).

**Immunofluorescence/Immunohistochemistry**. Mouse tumors were fixed in zinc fixative overnight prior to tissue processing and paraffin embedding followed by sectioning cut at 5 μm for immunofluorescent staining. All primary antibodies were sequentially stained for 1 h at RT diluted in a blocking/diluent buffer (Perkin Elmer) at the following concentrations: anti-mouse CD8 (1:200, 4SM15), and anti-phosphoSMAD2 (1:10,000, EPR2856) followed by MACH-2 anti-Rabbit or Mouse HRP-conjugated polymer (Biocare Medical) or for 10 min. at RT or anti-Rat/HRP polymer (Vector labs) for 30 min. at RT. Fluorescent signal was produced by staining with TSA-conjugated Opal dyes (Perkin Elmer) for 10 min. at RT using OPAL-520, OPAL-540, and OPAL-620. Nuclei were counterstained with DAPI. Whole tissues were scanned at 10× and at least 9 regions of interest per tissue section were imaged at 20× using Vectra 3.0 spectral imaging system (PerkinElmer).

**Flow cytometry**. For ex vivo cytokine analysis, cells were first treated with a 1× cell activation cocktail of PMA/ionomycin/Brefeldin A (Biolegend for 5 h in complete RPMI media. $1 \times 10^6$ cells from single cell suspensions were stained with anti-CD16/CD32 Fc block (1:200, BD Biosciences) and fixable viability 700 dye (1:10,000, BD Biosciences) in 1× PBS for 15 min. at 37 C prior to surface and intracellular staining with primary antibodies. Surface staining commenced in 200 μl FACS buffer supplemented with a 1:4 dilution of Brilliant Violet stain buffer (BD Biosciences) and fluorescently conjugated antibodies from the table below for 30 min. at 4 C in the dark; all are mouse reactive antibodies unless otherwise indicated. Following surface staining cells were washed and fixed in either 2% PFA or Fix/Perm buffer (eBioscience) for 20 min. at 4 °C for intracellular stain. Fix/perm buffer was washed with 1× perm wash buffer (eBioscience) and intracellular proteins were stained with fluorescently conjugated ICS antibodies in perm wash buffer for 30 min at 4 °C in the dark. Cells were washed and resuspended in 1× PBS prior to acquisition on a BD Fortessa or LSRII flow cytometer (BD Biosciences). Please refer to Supplementary Table 1 for a complete list of antibodies used.

**Chromatin Immunoprecipitation (ChIP)**. Mouse primary CD8+ T cells were isolated and purified from spleens by magnetic negative selection using a mouse CD8α + T cell isolation kit (Miltneyi Biotec). Cells were then plated in complete media on αCD3/αCD28 (1 μg ml$^{-1}$) coated 6 well plates at $2 \times 10^6$ cells/well to initiate rapid expansion. Seventy-two hours later, cells were harvested and plated in T-75 flasks in fresh complete RPMI media supplemented with 60 units/ml human IL-2. Media was exchanged with new IL-2-containing media every 48 h thereafter until cultures reach $>180 \times 10^6$ cells. Cells were harvested and exchanged with serum starvation media (complete RPMI media + 0.2% FBS) overnight and then stimulated with or without 2 ng/ml mouse recombinant TGFβ1 (R&D systems) for 1.5 h. Jurkat cells were cultured and serum starved with 0.2% FBS complete RPMI media for overnight and stimulated with or without 10 ng/ml human recombinant TGFβ1 (PeproTech) for 1.5 h. $4 \times 10^7$ Jurkat cells were used for each immunoprecipitation and $1.5 \times 10^7$ mouse CD8+ T cells were used for one immunoprecipitation. ChIP was performed following SimpleChIP Enzymatic Chromatin IP Kit (Cell Signaling Technology) per manufacturer's instructions. Briefly, chromatin from fixed cells was sonicated by Vibra-cell VC130 (Sonics&Materials) for 2 cycles of 6 s ON and 30 s OFF at 6–9 output watt. Chromatin was incubated with anti-rabbit IgG (1:500), anti-SMAD2 (1:50), or anti-SMAD3 (1:50) (Cell Signaling Technology) at 4 °C for overnight with rotation. Immunoprecipitated samples were

eluted and the DNA cross-links were reversed at 65 °C for 5 h or overnight. Sheared chromosomal DNA was subjected to quantitative RT-PCR using FastStart Universal SYBR Green Master (Roche) and StepOnePlus Real-Time PCR system (Thermo Scientific). The amount of immunoprecipitated DNA is represented as signal relative to the total amount of chromatin used as input. Data is shown as a fold change relative to a control rabbit IgG sample. Primer sequences (5′ to 3′) used were as follows: human CXCR3 FOR: AAGCTGGGCCTGATTCTGTC, REV: AAGTCTGTGGTGGGCTTCTG. mouse CXCR3 FOR: GGCTCCTCCTGACAAC AGAC, REV: TGCCCAGGCTGACTTCATAC.

**T cell adoptive co-transfer experiments**. CD8$^+$ T cells were purified from spleens of naïve CD45.1 C57BL/6 and CD45.2 ALK5$^{\Delta CD8}$ mice, mixed in equal ratios and labeled with 1 μM CFSE (Molecular Probes) prior to adoptive transfer into C57BL/6 mice bearing MC38 tumors on day 14 post-implant. Seven days following transfer, tumors, spleens and draining lymph nodes were harvested for FACS analysis.

**Cytotoxicity assay**. C57BL/6 and ALK5$^{\Delta CD8}$ mice were vaccinated with a replication-deficient Listeria monocytogenes vaccine vector engineered to express ovalbumin (ΔActA-OVA). Mice were primed with 10$^7$ bacteria intravenously followed by a boost 3 weeks later. Seven days following the vaccine boost CD8$^+$ T cells were purified from spleens via magnetic negative selection and labeled with 10 μM CFSE prior to co-culture with various ratios of unlabeled MCA205-OVA or Panc02 tumor cells. Realtime co-cultures were monitored with the IncuCyte S3 Live-Cell Analysis system (Sartorius) in the presence of Cytotox Red reagent (Essen Biosciences) for the detection of dead cells. The percent specific tumor cell cytotoxicity was calculated as follows [(total dead cells − dead CFSE$^+$ T cells)/total number of tumor cells plated] × 100.

**In vitro T cell activation**. CD8$^+$ T cells were purified from spleens of naïve C57BL/6 and ALK5$^{\Delta CD8}$ mice via magnetic bead negative selection and labeled with 1 μM CFSE prior to culture. Cells were cultured at $1 \times 10^5$ cells/well on αCD3/ αCD28 (1 μg ml$^{-1}$, 10 μg ml$^{-1}$) coated 96 well plate in complete RPMI medium. Following 72 h of culture, cells were harvested for FACS analysis of CFSE dye dilution and analyzed using the proliferation plug-in on FlowJo software (BD Biosciences). Supernatants were also collected from these cultures for cytokine analysis by cytokine bead array using the mouse inflammation kit (BD Biosciences) per manufacturer's instructions. In some assays, certain groups received recombinant mouse TGFβ1 (R&D systems) at 1 ng/ml at the initiation of the experiment.

**T cell migration assay**. CD8$^+$ T cells were purified from spleens of naïve CD45.1 C57BL/6 WT and CD45.2 ALK5$^{\Delta CD8}$ mice and mixed in equal ratios of $1 \times 10^5$ cells/genotype prior to plating in the top well of a 96 well transwell plate with a 3 μm pore size. Complete RPMI media with or without increasing concentrations of CXCL10 or CXCL16 was placed in the bottom chamber of the transwell plate. Cells were collected from the bottom portion of the well following 24 h culture and analyzed by FACS for the number of WT or KO T cells.

**PCR for Cre excision of ALK5**. CD4$^+$ T cells, B cells, CD8$^+$ T cells, Foxp3$^+$ Tregs, and macrophages were FACS sorted with an Aria II (BD Biosciences) and collected directly into cell lysis solution for genomic DNA (gDNA) isolation. gDNA was subsequently isolated using a DNeasy blood and tissue kit (Qiagen). PCR was performed with a 3-primer system using Terra PCR direct polymerase mix (Clontech) at a 58 degrees C annealing temperature with the following primers: ALK5 F (lnl5′) ATGAGTTATTAGAAGTTGTT, ALK5WT R (lnl3′) ACCCTCTCACTCTTCCTGAGT, and ALK5KO R (llox3′) GGAACTGG- GAAAGGAGATAAC[30]. PCR products were electrophoresed on a 1.5% agarose gel stained with GelRed DNA stain.

**Luminex analysis**. Tumor cytokine amounts were determined by a Mouse cytokine/chemokine procarta 36-plex Luminex kit (Invitrogen). MC38 tumors were harvested at day 14 post-implant and immediately snap frozen. Tumor lysates were then isolated by mechanical bead agitation in PBS + protease/phosphatase inhibitors and frozen for subsequent Luminex analysis. Protein amounts were calculated at total pg analyte/total mg protein as determined by BCA assay (Thermo Fisher).

**TCGA data**. RNAseq data was mined from TCGA—colorectal PanCancer Atlas data set on the cBioPortal for cancer genomics (cbioportal.org). SMAD2 and CXCR3 normalized mRNA expression levels were compared by linear regression analysis.

**Statistics and data analysis**. Graphpad Prism 7.0 software was used to construct all graphs and calculate statistical significance. FlowJo software was used for FACS analysis and to generate tSNE plots. When comparing two groups within an experiment the unpaired, two-sided student's $t$-test was used to determine statistical significance. When more than two biological or treatment groups were

compared, ANOVA was used to calculate p values. Significance from Kaplan-Meier survival curves were calculated with the Log-Rank test. Chi-squared or Fischer's exact test were used to compare rates of tumor rejection between groups where indicated.

**Reporting summary**. Further information on research design is available in the Nature Research Reporting Summary linked to this article.

## Data availability

The data is available in the Article and Supplementary Information or available from the corresponding author upon reasonable request. TCGA Colorectal Adenocarcinoma PanCancer Atlas data for CXCR3 expression was queried from: https://www.cbioportal.org/results/coexpression?Action=Submit&RPPA_SCORE_THRESHOLD=2.0&Z_SCORE_THRESHOLD=2.0&cancer_study_list=coadread_tcga_pan_can_atlas_2018&case_set_id=coadread_tcga_pan_can_atlas_2018_rna_seq_v2_mrna&data_priority=0&gene_list=CXCR3&geneset_list=%20&genetic_profile_ids_PROFILE_MRNA_EXPRESSION=coadread_tcga_pan_can_atlas_2018_rna_seq_v2_mrna_median_Zscores&profileFilter=0&tab_index=tab_visualize.

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

## Acknowledgements

We would like to thank Dr. Walter Urba for his support, advice, and editorial feedback on this manuscript. Funding support for this work came from The Sidney Kimmel Foundation Translational Research Scholar Award to K.H.Y., Providence Opportunity Fund Foundation Grant to K.H.Y., and Collins Medical Trust Grant to A.J.G. and K.H.Y.

## Author contributions

Supervision: A.J.G., T.Y., M.J.G., M.R.C., K.H.Y. Experimental design: A.J.G., T.Y., K.M., N.F., M.P., A.A., T.B., M.W., T.C., M.J.G., M.R.C., K.H.Y. Data acquisition: A.J.G., T.Y., K.M., N.F., M.P., A.A., T.B., K.H.Y. Data analysis: A.J.G., T.Y., K.M., N.F., M.P., A.A.,

T.B., M.J.G., M.R.C., K.H.Y. Manuscript writing: A.J.G., T.Y., N.F., K.H.Y. Manuscript editing: A.J.G., T.Y., N.F., A.A., M.W., D.O., R.A., M.X.K., A.H., M.J.G., M.R.C., K.H.Y.

## Competing interests

LY2157299 and LY3200882 were used under MTA with Eli Lilly. The Earle A. Chiles Research Institute receives support from Bristol-Myers-Squibb. There is an ongoing clinical trial testing these reagents at the Earle A. Chiles Research Institute, which is partially funded by Eli Lilly through their ExIST program, for which K.H.Y. is the Principal Investigator. The remaining authors declare no competing interests.
