## [Peer Review File · Nature Communications]

Editorial Note: Parts of the author's response and the reviewer's comments have been redacted as they refer to data that is no longer included in the paper.

Reviewers' comments:

Reviewer #1 (Remarks to the Author):

Over the last few years, several publications have shown the beneficial effect of using anti-TGF β to improve the response to cytotoxic therapy in cancer. Although it was shown that there is an association of anti-TGF β therapies with enhanced T cell infiltration and hence anti-tumor immunity, the mechanism by which TGF β achieves that is unknown. In this manuscript, the authors explored the possible mechanisms by which TGF β contributes to tumor immune suppression. They demonstrate a novel mechanism by which TGF β contributes to tumor immunosuppression through downregulation of CD8+ T cell expression of CXCR3 which limits the T cells trafficking to the tumor in a mouse model of rectal adenocarcinoma. The authors found an inverse correlation between SMAD2 and CXCR3 expression tumors after ALK5 inhibition after analyzing the TCGA colorectal database. Samples obtained from patients enrolled in an ongoing clinical trial at the author's institution of LY2157299 prior to standard of care chemoradiation in locally advanced rectal cancer shows that the majority of patients on study had a decrease percentage CXCR3+CD8+ T cells in peripheral blood but there was an increase in the percentage CXCR3+CD8+ T cells in the tumor microenvironment after LY2157299 treatment. This an interesting mechanistic study that sheds light on the effect of TGF β on tumor immunity. The study is well conducted and the experimental design seems appropriate. The statistical analysis is applied appropriately.

Concerns:

1. The authors discussed the use of TGF β inhibitor, the LY157299 as well as another more ALK5 specific drug, the LY3200882. The author seemed to be using the less specific drug in their animal experiment despite the availability of the more specific drug please explain.
2. The authors reported that they observed a non-significant increase in cure rates among ALK5 Δ CD8 animals receiving radiation, 50% versus 13.6% in WT mice (Fig. 2A-C). It looks like in figure 2-C iv that the effect of radiation of these mice is less than non-radiated mice in panel 2-Cii (9/15 vs 5/10) when compared to ALK5 Δ CD8 what the authors should be comparing to not the WT Please explain.
3. When showing data on tumor size in live mice, the authors show the tumor area in mm². Since the tumor growth in mice is a 3-dimensional structure the tumor volume should be assessed as volume not area. The tumor area X & Y dimension can be similar but the Z-axis can be different and hence this will reflect different aspects for comparison.

Minor Concerns:

1. No Statistics applied to figure 1B-i.

Reviewer #2 (Remarks to the Author):

The manuscript entitled "TGF β suppresses CD8+ T cell expression of CXCR3 and tumor trafficking" describes novel data showing that the multipotent cytokine TGF β subverts antitumor immunity by negatively targeting the chemokine receptor CXCR3 axis. This function is primarily targeted to CD8+ T cells and results in their exclusion from the tumor microenvironment. The authors show, using multiple approaches, that interruption of TGF β signaling increases effector T cell accumulation in tumors, and improves survival in mouse tumor models. Importantly, the authors report preliminary results from a clinical trial combining an inhibitor of TGF β signaling in rectal cancer patients, showing increased accumulation of CXCR3+CD8+ T cells in tumors. Together, the manuscript provides a compelling body of data supporting a novel role of TGF β in inhibiting anti-tumor immune responses. However, more detailed information regarding CXCR3 biology should be

provided along with additional data to support the conclusion that interruption of TGF β signaling increases CXCR3 expression levels in CD8+ T cells.

Major comments:

- More in-depth context should be provided throughout the manuscript regarding what is already known about the role of CXCR3 in cancer immunity. Although CXCR3 is the main focus of the study, the Introduction does not describe CXCR3 biology while the discussion of CXCR3 is somewhat superficial in the Results and Discussion.
- For the Discussion, the authors should discuss their findings in the context of the requirement for CXCR3 in recruitment of CD8+ effector T cells into the tumor microenvironment as reported previously (e.g., Mikucki et al., Nature Comm 2015; Clancy-Thompson, et al., Cancer Immunology Research 2015; Sprangler et al., Cancer Cell 2017, Woods et al., Cancer Immunology Research 2017), as well as CXCR3 functions within the tumor interstitial space to promote anti-tumor effector functions (Chow et al, Immunity 2019). The data provided are consistent with either scenario, so both should be discussed, citing appropriate references. Similarly, published data documenting that radiation induces the expression of CXCR3 ligands should be more fully discussed (e.g., Lugade, et al., Journal of Immunology 2008; Lim et al., Cancer Immunology, Immunotherapy 2014; Vanpouille-Box et al, Clinical Cancer Research 2018).
- The authors repeatedly state that ALK5 deletion enhances CXCR3 expression, yet the data provided mainly evaluates this metric by showing changes in the percent CXCR3-positive T cells rather than directly measuring CXCR3 expression levels (e.g., Figures 4 E, 4F, 5C). The authors need to present a more convincing argument to support the conclusion that CXCR3 protein expression is altered in vivo and in vitro following disrupted TGF β /ALK5 signaling or treatment with recombinant TGF β (i.e., significant changes in MFI values and representative flow histograms should be shown).
- CD44-low is not the conventional phenotype for effector cells. The authors should include the gating strategy and data validating that CD44-low cells are bona-fide effector cells. Data should also be re-analyzed using the conventionally-recognized CD44-hi L-selectin-negative effector phenotype.
- Endogenous T cells should be evaluated for CXCR3 downregulation to complement adoptive transfer data.
- In several instances, data referred to as 'not shown' are crucial to support the exposition and should be shown; i.e., in lines 267 (intratumoral chemokine/cytokine levels, 284-285 (CXCL10 levels), 320 (loss of CXCR3+ T cells over time), and 346-7 (CXCR3 MFI in patients).
- The authors should include discussion of the anatomy of the CXCR3-dependent antitumor response; specifically, how TGF β produced in the tumor microenvironment is predicted to influence CXCR3 expression in draining lymph nodes and systemic immunity, as suggested by the data shown in Figure 4B, C, E.

Minor comments:

- The result regarding CD4 depletion in Figure 1C is as strong as any other intervention shown, so it shouldn't be dismissed as a slight effect.
- The "cure ratio" should be included in Figure 4I
- The markers used to define cell populations are not always identified (e.g., in Supplementary Figures 1B, 4C, 4E).
- On line 882: Supplementary Figure 3F should be corrected to D.
- The heat maps in Supl Fig 4A are not mentioned in the results.
- The total number of experiments performed should be indicated for all Supplementary figures.
- There are several instances of undefined acronyms.

Reviewer #3 (Remarks to the Author):

This paper evaluates the role of pharmacological targeting of the TGFbeta receptor (ALK5) and genetic targeting of the ALK5 receptor in CD8 T-lymphocytes in tumor response in combination with standard of care chemoradiotherapy. Syngeneic CT26 and MC38 colorectal subcutaneous tumor models are used. Data from an ongoing clinical trial are included using biopsy material before and after one cycle of an ALK5 inhibitor. Clinical outcome of this study is not yet known. The findings of this paper are novel and contribute to our understanding why CD8 T lymphocytes poorly penetrate into tumors. The results may lead to novel therapies (the LY inhibitor used in the study and already in clinical trial) in combination with standard of care therapy or immunotherapy.

The title is misleading, since no evidence is shown that TGFbeta is responsible for observed effects. More correct would be that the TGFbetaReceptor and more specifically ALK5 is suppressing CXCR3 expression in CD8 T-cells influencing tumor trafficking.

Please describe the specificity of the LY2157299; comparison between TGFbetaRI vs II; other ALK family members? This is currently unclear and should be mentioned in this manuscript. On what basis is the dose of 150mg/kg LY inhibitor chosen? Was this experimentally validated in previous experiments? What are the mouse plasma levels and do the mouse plasma levels correlate with the human plasma levels?

Figure 1: Why are measurements continued until day 30 in triple treatment and stopped at day +/-23 for RT+5FU (and all others). At day 30 few of the animals reached humane endpoint and it is unclear why the area measurements stopped. Indicate animals at risk in Kaplan Meier survival plots.

A general comment of high importance is that all the experimental xenograft models are subcutaneous. It is well known that stromal content (such as cancer-associated fibroblasts and blood vessel organization) in subcutaneous tumors is different compared to orthotopic tumors and that this may cause changes in the response to ALK5 inhibitors and other therapies used in this study. One of the colorectal models (eg the CT26) subcutaneous model should be complemented by an orthotopic model such as in the caecum. The response observed might be dependent upon the ectopic injection site and this needs to be properly addressed.

Figure 2C: RT seems not to augment the impact of ALK5 knock-out in the CD8 T-lymphocytes. In contrast, Kaplan-Meier curve is slightly worse if RT is combined with ALK5 deletion in CD8 T-lymphocytes. Histology should be shown of mice "cured". Are any residual tumor cells left? Is a fibrotic core formed?

In all individual tumor growth curve experiments it is clear that some animals show an effective response and others don't. Not the slightest impact on growth is observed. What would be the reason of this therapy escape?? Where the levels of CXCR3 receptor in CD8 Tcell analysed in those mice? Are these differences experiment dependent? Or was this equally observed in a second experiment?

Figure 4I) please indicate number of mice that show response/to total number in growth curves as is shown in figure 2A-D.

Patient data are interesting, especially the immunocytochemistry on biopsies before treatment and after first cycle of LY inhibitor. In most patients it is clear that a higher infiltration of CXCR3+ CD8-T lymphocytes in tumors is observed after first cycle of LY treatment. Two important questions: 1) there is no control group receiving the standard of care treatment? 2) the image of day 1 versus day 15 is somewhat misleading. A clear epithelial differentiation of the tumor is shown in image of day 15 which is not the case in image of day 1. Preferentially patient with biopsy samples available

showing both epithelial structures should be shown. Currently, spatial differences and consequently stromal differences may be an important confounder in the current findings. By coincidence the image of day 1 may contain a majority of CAF showing less CXCR3+ CD8-T-cell infiltration versus a biopsy at day 15 with less CAF abundance and consequently more CXCR3+ CD8 T cells. Analysis of CAF markers and/or epithelial markers for their relative abundance could assist.

Reviewer #4 (Remarks to the Author):

This is an interesting and thoughtful piece of work that focuses on addressing the mechanism by which TGF β inhibition might improve the efficacy of rectal chemoradiation. It is a largely well written manuscript. However, some figures were crammed in an attempt to fit the page which made the review process slightly bothersome. The authors use appropriate murine models to establish enhanced responses to TGF β inhibition in combination with rectal chemoradiation and provide convincing evidence that CD8 α + T cells are a target of TGF β small molecule inhibition. It was encouraging to see the attempts made to correlate pre-clinical findings with tissue obtained from their prospective clinical study assessing the benefit of the addition of TGF β inhibition to rectal chemoradiation, thereby enhancing the clinical relevance of this research. However, there are some issues that need to be addressed that are outlined below:

1. For clarity, the authors should include a figure illustrating the proposed mechanism by which TGF β inhibits CXCR3 expression on CD8+ T cells and the impact of this on the threshold for TCR activation and their trafficking into tumours.
2. Short course radiotherapy (25Gy in 5 fractions daily over 1 week) is indeed used in the neoadjuvant setting for the treatment of some patients with rectal cancer but not in combination with 5FU or other cytotoxic drugs. Therefore, it is not true to say that the dosing schedule in the pre-clinical study mirrors standard of care.
3. Figure 1. Please indicate the number of mice in each treatment arm and the statistical test used. It is important to understand the variance of tumour sizes within each arm or illustrate individual data points for each animal.
4. Please show the survival data for the RT+LY treatment arm.
5. For purposes of comparison, it would be helpful to see data for α CD4+RT+LY treatment arm.
6. The figure key is missing from figure 1C.
7. Please indicate why such large doses of radiotherapy were used (10Gy x2) in experiments illustrated in figure 2. Were smaller, clinically relevant doses also assessed?
8. In figure 2D(iv), please explain why only 4 animals were used in this important treatment arm.
9. For the clinical component of this study, the decrease in peripheral CXCR3 expression is not convincing and should not be overstated. Ideally, a control arm (CRT with 5FU/capecitabine without LY) should be compared in figure 5D.
10. The statistical tests for all comparisons in this study should be stated clearly in the main body of the manuscript as well as the figure legends.
11. For all the figure legends, the relevant findings should be documented, not only the experimental detail.

Reviewers' comments:

Reviewer #1 (Remarks to the Author):

Concerns:

1. The authors discussed the use of TGF β inhibitor, the LY157299 as well as another more ALK5 specific drug, the LY3200882. The author seemed to be using the less specific drug in their animal experiment despite the availability of the more specific drug please explain.

This reviewer makes an excellent point. Our reasons are two-fold: 1) this study was initially designed to mirror our clinical trial which utilizes the LY2157299 compound, and 2) we did not have access to the more specific LY3200882 compound (through MTA) until the end of this particular study, when the manufacturer no longer made the LY2157299 inhibitor. Since the vast majority of the data reported here had already been compiled, we felt the best strategy for both compounds was not to repeat all of the results with the LY3200882 inhibitor but to verify it had similar or improved efficacy to the LY2157299, as we have reported in this manuscript and finish the few experiments remaining with the new drug. Furthermore, repeating all of these studies with the new inhibitor would have been outside the agreed upon scope of work for this compound under our MTA. As the drug company has decided to move forward clinically with the LY3200882 drug, we feel demonstrating similar effects with two different inhibitors with differing TGF β R1 selectivity strengthens our claims these small molecules are predominantly effecting the same pathway. We have clarified this in the discussion of the manuscript.

2. The authors reported that they observed a non-significant increase in cure rates among ALK5 Δ CD8 animals receiving radiation, 50% versus 13.6% in WT mice (Fig. 2A-C). It looks like in figure 2-C iv that the effect of radiation of these mice is less than non-radiated mice in panel 2-Cii (9/15 vs 5/10) when compared to ALK5 Δ CD8 what the authors should be comparing to not the WT Please explain.

We apologize for any confusion. Our intent in this figure was to show increased response to radiation in the ALK5 Δ CD8 mice compared to WT mice, thus the statistics compared WT+RT to ALK5 Δ CD8 mice + RT. We do make note of this in the results section (lines 140-142) but have added additional language to clarify this point. The tumor rejection rate in ALK5 Δ CD8 mice was so high, that in order to treat enough ALK5 Δ CD8 mice with RT we had to select individual mice that escaped tumor rejection – therefore few ALK5 Δ CD8 mice would have been cured in the group that had tumors at day 15 (when RT was delivered) but did NOT receive radiation. However, to further clarify that there was a benefit to radiation in the KO tumors that did not reject, we have generated a Kaplan Meier curve comparing ALK5 Δ CD8 mice vs ALK5 Δ CD8 mice + RT limited to tumors that were present on day 15 when radiation was delivered (see below). This is now part of Fig 2C.

3. When showing data on tumor size in live mice, the authors show the tumor area in mm². Since the tumor growth in mice is a 3-dimensional structure the tumor volume should be assessed as volume not area. The tumor area X & Y dimension can be similar but the Z-axis can be different and hence this will reflect different aspects for comparison.

We appreciate this thoughtful commentary. We understand that Z-axis measurements may affect total tumor volume, however in our experience, measuring tumor area or volume has not significantly changed the relative differences between mice within each study. Furthermore, since it can be difficult to estimate the depth of the tumor below the skin surface and invading into fat and muscle tissue, we chose not to introduce that potential variability into our measurements. Additionally, many studies which report volumetric measures of subcutaneous tumors utilize formulas based on length and width measures alone, such as length x width² x

0.52, which assume spherical tumor growth and do not add additional information beyond our measures of tumor area.

Mino Concerns:

1. No Statistics applied to figure 1B-i.

Thank you for pointing this oversight. We have revised the figure to include the statistical differences between RT+5FU+LY and RT+5FU treatment groups.

Reviewer #2 (Remarks to the Author):

Major comments:

• More in-depth context should be provided throughout the manuscript regarding what is already known about the role of CXCR3 in cancer immunity. Although CXCR3 is the main focus of the study, the Introduction does not describe CXCR3 biology while the discussion of CXCR3 is somewhat superficial in the Results and Discussion.

We agree with the reviewer that the manuscript would be strengthened by more background on CXCR3. We have added a paragraph addressing the role of CXCR3 in tumor immunology relevant to the context of the paper.

• For the Discussion, the authors should discuss their findings in the context of the requirement for CXCR3 in recruitment of CD8+ effector T cells into the tumor microenvironment as reported previously (e.g., Mikucki et al., Nature Comm 2015; Clancy-Thompson, et al., Cancer Immunology Research 2015; Sprangler et al., Cancer Cell 2017, Woods et al., Cancer Immunology Research 2017), as well as CXCR3 functions within the tumor interstitial space to promote anti-tumor effector functions (Chow et al, Immunity 2019). The data provided are consistent with either scenario, so both should be discussed, citing appropriate references. Similarly, published data documenting that radiation induces the expression of CXCR3 ligands should be more fully discussed (e.g., Lugade, et al., Journal of Immunology 2008; Lim et al., Cancer Immunology, Immunotherapy 2014; Vanpouille-Box et al, Clinical Cancer Research 2018).

We appreciate these helpful suggestions for strengthening our discussion. We had included some of the references in the manuscript, but have added the others and expanded our discussion of them to highlight the breadth and background on previous CXCR3/CXCR3L studies.

• The authors repeatedly state that ALK5 deletion enhances CXCR3 expression, yet the data provided mainly evaluates this metric by showing changes in the percent CXCR3-positive T cells rather than directly measuring CXCR3 expression levels (e.g., Figures 4 E, 4F, 5C). The authors need to present a more convincing argument to support the conclusion that CXCR3 protein expression is altered in vivo and in vitro following disrupted TGF β /ALK5 signaling or treatment with recombinant TGF β (i.e., significant changes in MFI values and representative flow histograms should be shown).

We agree with the reviewer that we were unclear on this point and agree that the frequency of CXCR3+ T cells does not directly measure expression of CXCR3 protein on each individual T cell. The MFI of CXCR3 on these same cells is similar to the frequency of CXCR3+ cells in this case because CXCR3 expression was either positive or absent, thus the T cells expressing CXCR3 increases the median CXCR3 for the population. However, that data is provided below. The murine MFI data for Figs. 4E-F is shown in panels A-B, while the [redacted].

- CD44-low is not the conventional phenotype for effector cells. The authors should include the gating strategy and data validating that CD44-low cells are bona-fide effector cells. Data should also be re-analyzed using the conventionally-recognized CD44-hi L-selectin-negative effector phenotype.

We apologize if we did not make clear the gating strategy we used for CD8+ T cell subtypes. When we gate on CD44 we can clearly identify 3 levels of CD44 expression; high, intermediate, and negative (see gating below). CD44^{HI}/CD62L⁻ cells are routinely defined as “effector memory” cells as compared to CD44^{HI}/CD62L⁺ cells which are considered “central memory” (3). As is common, CD44⁻CD62L⁺ cells were denoted as “naïve”. The CD44^{INT}/CD62L⁻ population were termed “effector” cells, as has been previously reported by others(3,4). However, in anticipation of this question, we also further phenotyped the cells using markers of effector function, which is demonstrated in Fig. S4 and includes granzyme B, PD1, Ki67, and Ly6C. We have clarified this in the text to draw out this point out for the reader.

- Endogenous T cells should be evaluated for CXCR3 downregulation to complement adoptive transfer data. We did indeed evaluate the endogenous T cells in these experiments for CXCR3 expression and it was similar in expression to the transferred WT cells, therefore we concluded endogenous CXCR3 expression was not altered by adoptive transfer. Furthermore, the results in Supp. Fig. 5E are more indicative of the CXCR3 comparison between untreated/WT and TGFβ1 inhibited T cells in the endogenous setting.

- In several instances, data referred to as ‘not shown’ are crucial to support the exposition and should be shown; i.e., in lines 267 (intratumoral chemokine/cytokine levels, 284-285 (CXCL10 levels), 320 (loss of CXCR3+ T cells over time) [redacted].

We appreciate the thoroughness of this reviewer. Ideally we could provide all data, however there are practical limitations to what is feasible. Per this request, we have now included the cytokine analysis from WT and KO tumors in Supp Fig 4. However, we believe the loss of CXCR3 in exhausted cells has already been published (5), and can therefore be omitted. That data is included below for the reviewer/editor. [redacted].

- The authors should include discussion of the anatomy of the CXCR3-dependent antitumor response; specifically, how TGFβ produced in the tumor microenvironment is predicted to influence CXCR3 expression in draining lymph nodes and systemic immunity, as suggested by the data shown in Figure 4B, C, E.

Thank you for this suggestion. We have now elaborated on these points in the discussion and included a schema in Fig. 5F. We speculate based on the results of increased CXCR3 recruitment, CD4 depletion, and change in effector phenotypes that TGFβ both in the lymph node and in the tumor microenvironment is important for the treatment effect seen.

Minor

comments:

- The result regarding CD4 depletion in Figure 1C is as strong as any other intervention shown, so it shouldn't be dismissed as a slight effect.

We agree with the reviewer. In fact, we make significant note of this result in the first paragraph of the results section. Furthermore, we experimentally followed up on this point by addressing the expression level of TGFβ1/LAP in tumors of these mice pre and post-RT specifically because we hypothesized that Tregs were the main source of TGFβ1 to CD8+ T cells (Supplementary Fig. 1B-C). There was a word that may have been interpreted as dismissive in the discussion that was removed in the revised manuscript.

- The "cure ratio" should be included in Figure 4I

We have now included this information in the figure.

- The markers used to define cell populations are not always identified (e.g., in Supplementary Figures 1B, 4C, 4E).

We have added markers used in the respective figure legends as suggested.

- On line 882: Supplementary Figure 3F should be corrected to D.

We have made this correction, thank you.

- The heat maps in Supl Fig 4A are not mentioned in the results.

We have added language that references this result and their meaning. This reflects back to this reviewers concern regarding the effector phenotyping.

- The total number of experiments performed should be indicated for all Supplementary figures.

We apologize for this omission and have update the legends to include this important information.

- There are several instances of undefined acronyms.

We have combed through the manuscript to identify the acronyms in question. We have the made the necessary corrections as we identified them but if we have missed any in particular, please let us know. We apologize for the oversight.

Reviewer #3 (Remarks to the Author):

1. The title is misleading, since no evidence is shown that TGFbeta is responsible for observed effects. More correct would be that the TGFbetaReceptor and more specifically ALK5 is suppressing CXCR3 expression in CD8 T-cells influencing tumor trafficking.

We apologize if this reviewer felt misled, that was certainly not our intention. While the models we utilize are inhibitors or knockouts of the TGFβ type I receptor, we do test TGFβ1 function on the WT and knockout T cells in vitro, and perform ChIP with the downstream signaling mediators of TGFβ, Smad2 and Smad3. Together these implicate TGFβ as the molecule responsible for the downstream signaling effects observed. As Alk5 does not have a DNA binding role, we believe that changing to title to suggest that Alk5 is suppressive would lead to more confusion. To that end, we have not modified the title. If the editor feels that a different title is more appropriate, we would be happy to entertain alternatives.

2. Please describe the specificity of the LY2157299; comparison between TGFbetaRI vs II; other ALK family

members? This is currently unclear and should be mentioned in this manuscript. On what basis is the dose of 150mg/kg LY inhibitor chosen? Was this experimentally validated in previous experiments? What are the mouse plasma levels and do the mouse plasma levels correlate with the human plasma levels?

Thank you for this feedback. We had previously included a discussion of the lower selectivity of LY2157299 in lines 158-174 for ALK4 vs. LY3200882. Although LY2157299 has higher binding strength for ALK4 vs ALK5, we do not see additional efficacy in LY2157299 treated CD8-ALK5KO tumor bearing animals, suggesting the predominant therapeutic effects of this compound are for ALK5 signaling. We chose the dose of 150 mg/kg based on our preliminary experiments titrating the oral dose of this compound, and discussions with the manufacturer. These results are also published by others elsewhere(6). Regarding plasma levels in mice, we refer you *Herbertz S et al, Drug Design, Development and Therapy, 2015* for information on animal PK(7).

3. Figure 1: Why are measurements continued until day 30 in triple treatment and stopped at day +/-23 for RT+5FU (and all others). At day 30 few of the animals reached humane endpoint and it is unclear why the area measurements stopped. Indicate animals at risk in Kaplan Meier survival plots.

We apologize for any confusion. The measurements for mean tumor area can only be displayed until the first animal reaches its endpoint. Continued display of measures would be misleading after that time as the largest tumor would no longer be measured (as that animal would have been euthanized), so the average size would drop significantly –example below:

For a group of 6 tumor-bearing animals at day 23: 50, 45, 65, 85, 100, 75, 80 = mean 71

If the largest tumor reaches endpoint at day 25, then at day 27: 75, 60, 90, 100, 95, 95 = mean 85 whereas in reality had that tumor continued growing (approx. size 125), then the mean would be 91. Therefore, all measurements are censored once a single animal in that group reaches the endpoint. As such, the last day displayed will vary by group depending on when the first animal is euthanized. Graphs of individual growth curves are included to aid the reader in assessing the responses of each mice in each treatment group. Inclusion of the number at risk, while common in human trials, is not common in animal studies. Therefore to avoid confusing the reader, we propose to include the number at risk data below, but not include in the final manuscript.

For the revised Figure 1B:

Day	Vehicle	RT B + 5FU	RT B + 5FU + LY
0	19	15	20
21	19	15	20
23	17	14	19
25	7	13	18
28	5	7	16
30	4		
32			14
35			
37	1		13
39			
41		4	
43			
45		3	10
46			
47			
48		2	
49			
51			
52			9
54		1	7
57			
59			6
61			
75			5
82			4

Hazard Ratio (Mantel-Haenszel) RT B + 5FU/RT B + 5FU + LY	RT B + 5FU + LY	RT B + 5FU
Ratio (and its reciprocal)	3.021	0.331
95% CI of ratio	1.238 to 7.369	0.1357 to 0.8074

For Figure 1C:

Days	NT	RT A + 5FU	RT A + 5FU + LY	aCD8	aCD8 + RT A + 5FU + LY	aCD8 + RT A + 5FU	aCD4	aCD4 + RT A + 5FU	aCD4 + RT A + 5FU + LY
0	7	6	6	7	7	7	7	5	4
21	7	6		7	7	7	7		
23	3	5		2	3		6		
25	1	4			2		3		
28		3							
30			6				2		
32								5	4
35		2				3			
37									
41						2		4	
43			5					3	3
48						1			
50								2	
59			3						
62		1							
95			1					1	1

Hazard Ratio (Mantel-Haenszel) RT A + 5FU + LY/ aCD8 + RT A + 5FU + LY	reciprocal
Ratio (and its reciprocal)	0.04448
95% CI of ratio	0.007916 to 0.25
	22.48
	4 to 126.3

4. A general comment of high importance is that all the experimental xenograft models are subcutaneous. It is well known that stromal content (such as cancer-associated fibroblasts and blood vessel organization) in subcutaneous tumors is different compared to orthotopic tumors and that this may cause changes in the response to ALK5 inhibitors and other therapies used in this study. One of the colorectal models (eg the CT26) subcutaneous model should be complemented by an orthotopic model such as in the caecum. The response observed might be dependent upon the ectopic injection site and this needs to be properly addressed.

We appreciate that tumor location can influence tumor growth kinetics and the microenvironment. We would like to be clear that we have not used any xenograft models which comprise tumor implants into an immunocompromised animal. In this study all animal models were immune-competent. If the reviewers meant to highlight the difference between orthotopic and subcutaneous tumor models, we did consider this. However, we found the growth of orthotopic CT26 and MC38 too rapid to undergo treatment as we typically wait at least 7 days post-implant before initiating therapy to allow for tumor establishment and the development of an anergic/tolerogenic microenvironment. In our hands, the orthotopic tumors grew to endstage within 10-14 days, even at very low tumor cell inoculations, and were therefore not suitable for our studies. However, considering the inclusion of two different well-characterized colorectal mouse models in two different backgrounds as well as [redacted] and an independent TCGA colorectal cohort, we feel we have adequately tested our hypotheses across models/backgrounds/organisms.

5. Figure 2C: RT seems not to augment the impact of ALK5 knock-out in the CD8 T-lymphocytes. In contrary, Kaplan-Meier curve is slightly worse if RT is combined with ALK5 deletion in CD8 T-lymphocytes. Histology should be shown of mice "cured". Are any residual tumor cells left? Is a fibrotic core formed?

This comment is similar to Reviewer #1 comment 2 and is addressed above. Since there was no tumor left in cured mice at day 80 post-implant, we did not collect any tissue from the tumor injection site to assess for residual tumor cells, however there was nothing palpable at the injection site suggesting a fibrotic core.

6. In all individual tumor growth curve experiments it is clear that some animals show an effective response and others don't. Not the slightest impact on growth is observed. What would be the reason of this therapy escape?? Where the levels of CXCR3 receptor in CD8 Tcell analysed in those mice? Are these differences experiment dependent? Or was this equally observed in a second experiment?

The reviewer is correct that there exists a spectrum of response, with some mice CD8-ALK5KO mice rejecting tumors and others exhibiting tumor growth with variation from no effect to near rejection and then outgrowth.

We did not assess the levels of CXCR3 in these particular mice to know if there is a correlation with CXCR3 expression and tumor regression. In the co-transfer experiment CXCR3+CD8+ T cells were consistently increased in all the KO T cells (Figure 4E). However it has been published that exhausted CD8 T cells lose CXCR3 expression(5), and given the tumor outgrowth we would anticipate that the tumor-infiltrating CD8 T cells would have lost CXCR3 expression consistent with this and with our data in endogenous T cells shown above for Reviewer #2. All experiments were repeated 2-3 times, and results were very consistent. Results in Fig. 2C specifically is a combined analysis of 2 independent experiments. Variation in response to therapy was also observed in patients, as in Fig 5 demonstrating that most but not all patients had a decrease in blood CXCR3+ T cells after LY2157299 treatment corresponding to an increase in tumor CXCR3+CD8+ T cells. Finally, Fig. 4I demonstrates that CXCR3 neutralization, significantly but not completely reversed the tumor cure in CD8-ALK5KO mice indicating CXCR3 recruitment is one, but not the only mechanism responsible for tumor rejection in these mice. We make note of this in the revised discussion.

7. Figure 4I) please indicate number of mice that show response/to total number in growth curves as is shown in figure 2A-D.

We have added this information to be consistent with the rest of the manuscript.

8. [redacted]

[redacted]

Reviewer #4 (Remarks to the Author):

1. For clarity, the authors should include a figure illustrating the proposed mechanism by which TGF β inhibits CXCR3 expression on CD8+ T cells and the impact of this on the threshold for TCR activation and their trafficking into tumours.

We have included this figure in Fig. 5F.

2. Short course radiotherapy (25Gy in 5 fractions daily over 1 week) is indeed used in the neoadjuvant setting for the treatment of some patients with rectal cancer but not in combination with 5FU or other cytotoxic drugs. Therefore, it is not true to say that the dosing schedule in the pre-clinical study mirrors standard of care.

We appreciate the reviewer's accuracy in this regard. We performed studies of 2Gy x 15 fractions + 5-FU + LY and this was slightly more efficacious than 5Gy x 5 fractions + LY without chemotherapy. However, early presentation of our work raised questions as to how we could compare those groups without including a 5Gy x 5 + 5-FU +LY group, despite what is clinically delivered. Therefore, when we repeated the experiments, we included all treatment variations (12 groups). Both 2Gy x 15 + 5-FU + LY and 5Gy x 5 + 5-FU+LY performed equivalently (shown below, where RT A= 2Gy x 15 and RT B = 5Gy). For practical delivery, we opted for the shorter radiation course for our depletion studies. Based on this feedback, we have updated Figure 1 to display 2Gy x 15 + 5FU data in place of the 5Gy x 5 data, which is consistent with our clinical trial and with standard of care radiation.

3. Figure 1. Please indicate the number of mice in each treatment arm and the statistical test used. It is important to understand the variance of tumour sizes within each arm or illustrate individual data points for each animal.

We have now included the sample size for each treatment group in the figure and that statistical test in the figure legend.

4. Please show the survival data for the RT+LY treatment arm.

As we have updated Figure 1 to demonstrate the 2Gy x 15 fractionation scheme, we do not think this comment would be relevant as it was directed toward 5Gy x 5 + LY.

5. For purposes of comparison, it would be helpful to see data for α CD4+RT+LY treatment arm.

As we have updated Figure 1, this comment is again not relevant.

6. The figure key is missing from figure 1Ci

The legend for this panel is next to the survival curve. We have now made this clear in the legend.

7. Please indicate why such large doses of radiotherapy were used (10Gy x2) in experiments illustrated in figure 2. Were smaller, clinically relevant doses also assessed?

We apologize for any confusion on the radiation dosing schema in the MC38 model. We previously determined that 10Gy x 2 is the minimal effective dose of RT in the MC38 model capable of delaying tumor growth, but insufficient to cure animals. This dose was selected to allow for combination with additional therapy with the goal of improved survival. The use of the Cre-Flox knockout model systems was for mechanistic evaluation of TGF β R1 antagonism, but this artificial system does not replicate a clinical scenario, and therefore lower doses were not assessed.

8. In figure 2D(iv), please explain why only 4 animals were used in this important treatment arm.

This is only 1 experiment representative of 2 total studies. In both studies combined, 10 mice were used.

9. [redacted]

[redacted]

10. The statistical tests for all comparisons in this study should be stated clearly in the main body of the manuscript as well as the figure legends.

We apologize if this was not clear originally. We have added language and clarity on the statistics in multiple figure legends, particularly if a test was applied that was uncommon.

11. For all the figure legends, the relevant findings should be documented, not only the experimental detail.

We appreciate this suggestion. We have highlighted the important findings in the revised figure titles. Further elaboration of experimental findings can be found in the manuscript.

Thank you for your consideration of our re-revised manuscript.

References

1. Chakravarthy A, Khan L, Bensler NP, Bose P, De Carvalho DD. TGF- β -associated extracellular matrix genes link cancer-associated fibroblasts to immune evasion and immunotherapy failure. *Nat Commun* [Internet]. Springer US; 2018;9(1):4692. Available from: <http://www.nature.com/articles/s41467-018-06654-8>
2. Tauriello DVF, Palomo-ponce S, Iglesias M, Stork D, Sevillano M, Berenguer-Illego A, et al. TGF-beta drives immune evasion in genetically reconstituted colon cancer metastasis. *Nat Publ Gr. Nature Publishing Group*; 2018;(Imim).
3. Obar JJ, Lefrancois L. Memory CD8+ T cell differentiation. *Ann N Y Acad Sci*. 2010;1183:251–66.
4. Martin MD, Badovinac VP. Defining memory CD8 T cell. *Front Immunol*. 2018;9(NOV):1–10.
5. Subramaniam S, Haining WN, Barber DL, Blattman JN, Wherry EJ, Sarkar S, et al. Molecular Signature of CD8+ T Cell Exhaustion during Chronic Viral Infection. *Immunity*. 2007;27(4):670–84.
6. Holmgaard RB, Schaer DA, Li Y, Castaneda SP, Murphy MY, Xu X, et al. Targeting the TGF β pathway with galunisertib, a TGF β RI small molecule inhibitor, promotes anti-tumor immunity leading to durable, complete responses, as monotherapy and in combination with checkpoint blockade. *Journal for ImmunoTherapy of Cancer*; 2018;1–15.
7. Lahn M, Herbertz S, Sawyer JS, Stauber AJ, Gueorguieva I, Driscoll KE, et al. Clinical development of galunisertib (LY2157299 monohydrate), a small molecule inhibitor of transforming growth factor-beta signaling pathway. *Drug Des Devel Ther*. 2015;9:4479.
8. Young KH, Newell P, Cottam B, Friedman D, Savage T, Baird JR, et al. TGF β Inhibition Prior to Hypofractionated Radiation Enhances Efficacy in Preclinical Models. *Cancer Immunol Res* [Internet]. 2014 Jul 21 [cited 2014 Sep 4]; Available from: <http://www.ncbi.nlm.nih.gov/pubmed/25047233>

REVIEWERS' COMMENTS:

Reviewer #1 (Remarks to the Author):

The authors addressed all concerns appropriately. No further revision is required

Reviewer #2 (Remarks to the Author):

The manuscript documents the novel and broadly interesting finding that the multipotent cytokine TGF β subverts anti-tumor immunity by targeting the chemokine receptor CXCR3, resulting in CD8+ T cell exclusion of from the tumor microenvironment. The authors have largely responded to concerns of the prior review by providing additional context for their results in the revised manuscript. The authors have now appropriately cited relevant studies regarding the role of CXCR3 in antitumor immunity. It is recommended that the additional report by Spranger et al Cancer Cell 31:711, 2017) also be cited since this is an important publication in this field.

Additionally, the new data provided satisfactorily address most of the issues raised. However, while the data shown in the rebuttal letter are convincing and supportive of the manuscript, these data must be intercalated into the main figures or supplemental figures of the revised manuscript. This applies to this reviewer's prior Major Comments #3 (CXCR3 MFI), #5 (CXCR3 down-modulation on endogenous cells), and #6 (CXCR3 loss over time). Data should also be shown to document that CXCR3 loss is associated with T cell exhaustion; it would be appropriate to indicate that these latter findings are consistent with a cited published report. Moreover, the "cure ratio", which is shown in Figure 2 as "the number of mice cured over the total number of 768 tumor-bearing mice is shown in the top right of each graph", should be added to Figure 4I.

An additional point is that the newly added sentence starting on line 203 is incomplete and remains confusing. This section should be further revised to clearly define the phenotypes of both cell populations identified.

Reviewer #3 (Remarks to the Author):

The authors answered all questions sufficiently and is happy with the transparent reporting of some experimental data.

this reviewer, however, still has difficulties with the choice of TGFbeta as being responsible for the observed effects. the role of TGFbeta is only investigated in a minority of in vitro assays. the majority of data are obtained by using TGFbeta receptor inhibitors.

Nevertheless, since I'm the only reviewer that is questioning this aspect I agree with the authors of the study to stay with the existing title.

Reviewer #4 (Remarks to the Author):

Thank you for addressing my previous concerns.

REVIEWERS' COMMENTS:

Reviewer #1 (Remarks to the Author):

The authors addressed all concerns appropriately. No further revision is required

Reviewer #2 (Remarks to the Author):

The manuscript documents the novel and broadly interesting finding that the multipotent cytokine TGF β subverts anti-tumor immunity by targeting the chemokine receptor CXCR3, resulting in CD8+ T cell exclusion of from the tumor microenvironment. The authors have largely responded to concerns of the prior review by providing additional context for their results in the revised manuscript. The authors have now appropriately cited relevant studies regarding the role of CXCR3 in antitumor immunity. It is recommended that the additional report by Spranger et al Cancer Cell 31:711, 2017) also be cited since this is an important publication in this field.

We have added this important reference.

Additionally, the new data provided satisfactorily address most of the issues raised. However, while the data shown in the rebuttal letter are convincing and supportive of the manuscript, these data must be intercalated into the main figures or supplemental figures of the revised manuscript. This applies to this reviewer's prior Major Comments #3 (CXCR3 MFI), #5 (CXCR3 down-modulation on endogenous cells), and #6 (CXCR3 loss over time). Data should also be shown to document that CXCR3 loss is associated with T cell exhaustion; it would be appropriate to indicate that these latter findings are consistent with a cited published report.

These data are now included in Supplementary Figure 5.

Moreover, the "cure ratio", which is shown in Figure 2 as "the number of mice cured over the total number of 768 tumor-bearing mice is shown in the top right of each graph", should be added to Figure 4I.

We have reordered our figures. This is now included in Figure 5.

An additional point is that the newly added sentence starting on line 203 is incomplete and remains confusing. This section should be further revised to clearly define the phenotypes of both cell populations identified.

We apologize for the confusing wording. This sentence has been rewritten to clarify our intent.

Reviewer #3 (Remarks to the Author):

The authors answered all questions sufficiently and is happy with the transparent reporting of some experimental data. This reviewer, however, still has difficulties with the choice of TGFbeta as being responsible for the observed effects. the role of TGFbeta is only investigated in a minority of in vitro assays. the majority of data are obtained by using TGFbeta receptor inhibitors. Nevertheless, since I'm the only reviewer that is questioning this aspect I agree with the authors of the study to stay with the existing title.

Reviewer #4 (Remarks to the Author):

Thank you for addressing my previous concerns.

Thank you for your consideration of our re-revised manuscript.